# Certificate-Guided Pruning for Stochastic Lipschitz Optimization

Ibne Farabi Shihab [* 1]  Sanjeda Akter [* 1]  Anuj Sharma [2]

## Abstract

We study black-box optimization of Lipschitz functions under noisy evaluations. Existing adaptive discretization methods implicitly avoid suboptimal regions but do not provide explicit certificates of optimality or measurable progress guarantees. We introduce **Certificate-Guided Pruning (CGP)**, which maintains an explicit *active set* $A_t$ of potentially optimal points via confidence-adjusted Lipschitz envelopes. Any point outside $A_t$ is certifiably suboptimal with high probability, and under a margin condition with near-optimality dimension $\alpha$, we prove $\text{Vol}(A_t)$ shrinks at a controlled rate yielding sample complexity $\tilde{O}(\varepsilon^{-(2+\alpha)})$. We develop three extensions: CGP-Adaptive learns $L$ online with $O(\log T)$ overhead; CGP-TR scales to $d > 50$ via trust regions with local certificates; and CGP-Hybrid switches to GP refinement when local smoothness is detected. Experiments on 12 benchmarks ($d \in [2, 100]$) show CGP variants match or exceed strong baselines while providing principled stopping criteria via the computable gap proxy $\varepsilon_t$.

## 1. Introduction

Black-box optimization, the task of finding the maximum of a function $f : \mathcal{X} \to \mathbb{R}$ accessible only through noisy point evaluations, is fundamental to machine learning, with applications spanning hyperparameter tuning (Snoek et al., 2012; Bergstra & Bengio, 2012), neural architecture search (Zoph & Le, 2017), and simulation-based optimization (Fu, 2015; Brochu et al., 2010). In many such settings, evaluations are expensive: training a neural network or running a physical simulation may cost hours or dollars per query.

---
[*]Equal contribution [1]Department of Computer Science, Iowa State University, Ames, Iowa, USA [2]Department of Civil, Construction & Environmental Engineering, Iowa State University, Ames, Iowa, USA. Correspondence to: Ibne Farabi Shihab <ishihab@iastate.edu>.

*Proceedings of the 43ʳᵈ International Conference on Machine Learning*, Seoul, South Korea. PMLR 306, 2026. Copyright 2026 by the author(s).

We call these "precious calls" where each evaluation must count, motivating the need for methods that provide explicit progress guarantees. The Lipschitz continuity assumption provides a natural framework for addressing this challenge. If $|f(x) - f(y)| \le L \cdot d(x, y)$ for a known constant $L$, then observations at sampled points constrain $f$ globally, enabling pruning of provably suboptimal regions. Classical methods exploiting this structure include DIRECT (Jones et al., 1993; 1998), Lipschitz bandits (Kleinberg et al., 2008; Auer et al., 2002), and adaptive discretization algorithms (Bubeck et al., 2011; Valko et al., 2013; Munos, 2014). However, existing methods implicitly avoid suboptimal regions via tree-based refinement without exposing two properties that matter for precious call optimization: (1) explicit certificates identifying which regions are provably suboptimal at any time $t$, and (2) measurable progress indicating how much of the domain remains plausibly optimal.

To address these limitations, we introduce Certificate-Guided Pruning (CGP), which maintains an explicit active set $A_t \subseteq \mathcal{X}$ of potentially optimal points. This set is defined via a Lipschitz UCB envelope $U_t(x)$ that upper bounds $f(x)$ with high probability, a global lower certificate $\ell_t$ that lower bounds $f(x^*)$, and the active set $A_t = \{x : U_t(x) \ge \ell_t\}$. Points outside $A_t$ are certifiably suboptimal, and as sampling proceeds, $A_t$ shrinks, providing anytime valid progress certificates (for vanilla CGP with known or conservatively bounded $L$). Unlike prior work that uses similar mathematical tools implicitly, CGP exposes the pruning mechanism as a first-class algorithmic object: the certificate is computable in closed form, the shrinkage rate is provably controlled, and the certificate provides valid optimality bounds even when stopped early. Figure 1 illustrates this mechanism. To understand how CGP differs from existing approaches, consider zooming algorithms (Kleinberg et al., 2008; Bubeck et al., 2011). While zooming maintains a tree of "active arms" and expands nodes with high UCB, the implicit pruning is an analysis artifact not exposed to the user (Shihab et al., 2026). Table 1 makes this distinction precise: CGP provides explicit certificates, computable progress metrics, and principled stopping rules that zooming-based methods lack. Similarly, Thompson sampling (Thompson, 1933; Russo & Van Roy, 2014; Russo et al., 2018) and information-directed methods (Hennig & Schuler, 2012; Hernández-Lobato et al., 2014) maintain implicit uncertainty without providing ex-

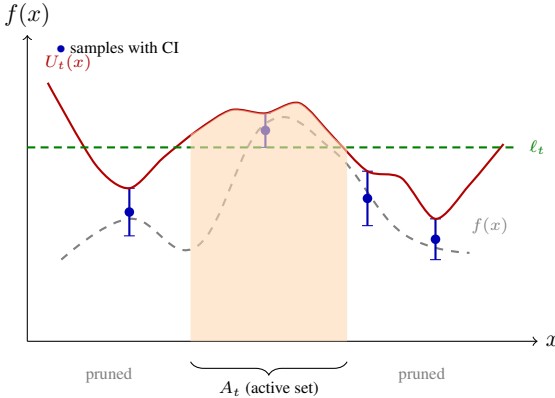

*Figure 1.* The active set $A_t$ (shaded) consists of points where the Lipschitz envelope $U_t(x)$ (red) exceeds the global lower bound $\ell_t$ (green dashed). Regions where $U_t(x) < \ell_t$ are certifiably suboptimal and pruned, causing $A_t$ to shrink as sampling proceeds.

*Table 1.* Comparison with zooming-based bandits. CGP uniquely exports explicit certificates (active set $A_t$, gap proxy $\varepsilon_t$, and volume $\text{Vol}(A_t)$), enabling principled stopping criteria that implicit zooming methods lack.

| Property | CGP | Zooming | HOO/StoSOO |
|---|---|---|---|
| Explicit active set $A_t$ | ✓ | – | – |
| Computable $\text{Vol}(A_t)$ | ✓ | – | – |
| Anytime optimality bound | ✓ | – | – |
| Principled stopping rule | ✓ | – | – |
| Certificate export | ✓ | – | – |
| Sample complexity | $\tilde{O}(\varepsilon^{-(2+\alpha)})$ | $\tilde{O}(\varepsilon^{-(2+\alpha)})$ | $\tilde{O}(\varepsilon^{-(2+d)})$ |
| Per-iteration cost | $O(N_t)$ | $O(\log N_t)$ | $O(\log N_t)$ |
| Adaptive $L$ | ✓(CGP-A) | – | – |
| High-dim scaling | ✓(CGP-TR) | – | – |

plicit geometric certificates.

Building on this foundation, our contributions are fourfold. First, we present CGP with explicit active set maintenance and prove a shrinkage theorem: under a margin condition with near-optimality dimension $\alpha$ (i.e., $\text{Vol}(\{x : f(x) \geq f^* - \varepsilon\}) \leq C\varepsilon^{d-\alpha}$), we show $\text{Vol}(A_t) \leq C \cdot (2(\beta_t + L\eta_t))^{d-\alpha}$, yielding sample complexity $T = \tilde{O}(\varepsilon^{-(2+\alpha)})$ that improves on the worst case $\tilde{O}(\varepsilon^{-(2+d)})$ when $\alpha < d$ (Section 4). We additionally provide an explicit algorithmic guarantee (Lemma 4.9) showing that Algorithm 1's actual mechanics drive $\beta_t$, $\eta_t$, and $\gamma_t$ to the required scales under an approximate-oracle assumption, addressing the connection from algorithm to rate. Second, we develop CGP-Adaptive (Section 5), which learns $L$ online via a doubling scheme, proving that unknown $L$ adds only $O(\log T)$ multiplicative overhead, the first such guarantee for Lipschitz optimization with certificates. Third, we introduce CGP-TR (Section 6), a trust region variant that scales to $d > 50$ by maintaining local certificates within adaptively sized regions, enabling high-dimensional applications previously intractable for Lipschitz methods. Fourth, we propose CGP-Hybrid (Section 7), which detects local smoothness

*Table 2.* Summary of Notation

| Symbol | Description |
|---|---|
| $f^*$ | Global maximum value of the objective function |
| $L$ | Lipschitz constant (or global upper bound) |
| $A_t$ | Active set at time $t$ (contains potential optimizers) |
| $U_t(x)$ | Lipschitz Upper Confidence Bound envelope |
| $\ell_t$ | Global lower certificate ($\max_i \text{LCB}_i$) |
| $\alpha$ | Near-optimality dimension (problem hardness) |
| $\beta_t$ | Active confidence radius (max $r_i$ over active points) |
| $\beta_{\text{sched}}(t)$ | Target confidence radius (replication schedule) |
| $\eta_t$ | Covering radius (resolution of $A_t$) |
| $\gamma_t$ | Gap to optimum proxy ($f^* - \ell_t$) |
| $\varepsilon_t$ | Computable gap proxy $2(\beta_t + L\eta_t) + \gamma_t$ |
| $\rho$ | Local smoothness ratio ($L_{\text{local}}/L_{\text{global}}$) |

via the ratio $\rho = L_{\text{local}}/L_{\text{global}}$ and switches to GP refinement when $\rho < 0.5$, achieving best of both worlds performance across diverse function classes. These theoretical contributions translate to strong empirical performance. Experiments (Section 9) demonstrate that CGP variants are competitive with strong baselines on 12 benchmarks spanning $d \in [2, 100]$, including Rover trajectory optimization ($d = 60$), neural architecture search ($d = 36$), and safe robotics where certificates enable stopping with guaranteed bounds (Shihab et al., 2025a). CGP-Hybrid performs best among tested methods on all 12 benchmarks under matched budgets, including Branin and Rosenbrock where vanilla CGP previously lost to GP-based methods.

## 2. Problem Formulation

Let $(\mathcal{X}, d)$ be a compact metric space with diameter $D = \sup_{x,y} d(x, y)$. We consider $\mathcal{X} = [0, 1]^d$ with Euclidean metric. Let $f : \mathcal{X} \to [0, 1]$ satisfy:

**Assumption 2.1** (Lipschitz continuity). There exists $L > 0$ such that for all $x, y \in \mathcal{X}$: $|f(x) - f(y)| \leq L \cdot d(x, y)$.

*Remark* 2.2 (No differentiability required). Assumption 2.1 is strictly weaker than differentiability: Lipschitz functions may have kinks, corners, and discontinuous gradients. This contrasts with GP-based methods, which implicitly assume smoothness via kernel choice (e.g., Matérn-$\nu$ with $\nu > 1$ requires differentiability). Our Needle-2D benchmark ($f(x) = 1 - \|x - x^*\|^{1/2}$, non-differentiable at $x^*$) illustrates this distinction empirically: CGP outperforms GP-UCB substantially (Table 3).

We observe $f$ through noisy queries: querying $x$ returns $y = f(x) + \epsilon$, where:

**Assumption 2.3** (Sub-Gaussian noise). The noise $\epsilon$ is $\sigma$-sub-Gaussian: $\mathbb{E}[e^{\lambda\epsilon}] \leq e^{\lambda^2\sigma^2/2}$ for all $\lambda \in \mathbb{R}$.

After $T$ samples, the algorithm outputs $\hat{x}_T \in \mathcal{X}$. The goal is to minimize simple regret $r_T = f(x^*) - f(\hat{x}_T)$. We seek PAC guarantees: $r_T \leq \varepsilon$ with probability $\geq 1 - \delta$.

**Assumption 2.4** (Margin / near-optimality dimension). There exist $C > 0$ and $\alpha \in [0, d]$ such that for all $\varepsilon > 0$: $\text{Vol}(\{x \in \mathcal{X} : f(x) \geq f^* - \varepsilon\}) \leq C\varepsilon^{d-\alpha}$.

The parameter $\alpha$ is the near-optimality dimension: smaller $\alpha$ means a sharper optimum (easier), $\alpha = d$ is worst case. For isolated maxima with nondegenerate Hessian, $\alpha = d/2$; for $f(x) \approx f^* - c\|x - x^*\|^p$, we have $\alpha = d/p$. This assumption is standard in bandit theory (Audibert & Bubeck, 2010; Bubeck et al., 2011; Lattimore & Szepesvári, 2020); see Appendix A for extended discussion.

## 3. Algorithm: Certificate-Guided Pruning

With the problem formalized, we now describe the CGP algorithm (Algorithm 1). CGP maintains sampled points with empirical estimates, confidence intervals, and the active set. At time $t$, let $\{x_1, \ldots, x_{N_t}\}$ be distinct points sampled, with $n_i$ observations at $x_i$. Define empirical mean $\hat{\mu}_i(t) = \frac{1}{n_i}\sum_{j=1}^{n_i} y_{i,j}$ and confidence radius $r_i(t) = \sigma\sqrt{2\log(2N_tT/\delta)/n_i}$, ensuring $|f(x_i) - \hat{\mu}_i(t)| \leq r_i(t)$ with high probability.

The upper and lower confidence bounds are $\text{UCB}_i(t) = \hat{\mu}_i(t) + r_i(t)$ and $\text{LCB}_i(t) = \hat{\mu}_i(t) - r_i(t)$. The global lower certificate $\ell_t = \max_{i \leq N_t} \text{LCB}_i(t)$ satisfies $\ell_t \leq f(x^*)$ under the good event. The Lipschitz UCB envelope propagates uncertainty:

$$U_t(x) = \min_{i \leq N_t} \{\text{UCB}_i(t) + L \cdot d(x, x_i)\}, \qquad (1)$$

which upper-bounds $f(x)$ everywhere. The active set is

$$A_t = \{x \in \mathcal{X} : U_t(x) \geq \ell_t\}, \qquad (2)$$

and points outside $A_t$ are certifiably suboptimal: their upper bound is below the lower bound on $f^*$.

The algorithm selects queries via $\text{score}(x) = U_t(x) - \lambda \cdot \min_{i \leq N_t} d(x, x_i)$ where $\lambda = L$, selecting $x_{t+1} = \arg\max_{x \in A_t} \text{score}(x)$. The first term favors high UCB regions while the second encourages coverage. CGP allocates additional samples to active points with $r_i(t) > \beta_{\text{sched}}(t)$ to reduce confidence radii. The target confidence radius follows a schedule $\beta_{\text{sched}}(t) = \sigma\sqrt{2\log(2T^2/\delta)/t}$, ensuring that confidence radii decrease at rate $O(1/\sqrt{t})$.

*Remark* 3.1 (Notation: $\beta_t$ vs. $\beta_{\text{sched}}(t)$). The two $\beta$ quantities play distinct roles: $\beta_t = \max_{i:x_i \text{ active}} r_i(t)$ is a *descriptive* quantity (the observed worst confidence over active points, used in the shrinkage theorem), while $\beta_{\text{sched}}(t)$ is the *prescribed schedule* that the replication rule enforces. The replication rule guarantees $\beta_t \leq \beta_{\text{sched}}(t)$ at all times.

We compute $A_t$ via discretization for low dimensions and Monte Carlo sampling for $d > 5$ (details in Appendix B.2). Theoretically, CGP assumes an approximate-oracle (Definition 4.8 below) for $\arg\max_{x \in A_t} \text{score}(x)$; practically,

---

**Algorithm 1** Certificate-Guided Pruning (CGP)

**Require:** Domain $\mathcal{X}$, Lipschitz constant $L$, noise $\sigma$, budget $T$, confidence $\delta$
1: Initialize: Sample $x_1$ uniformly, observe $y_1$
2: **for** $t = 1, \ldots, T - 1$ **do**
3:     Compute $r_i(t)$, $\ell_t = \max_i \text{LCB}_i(t)$, $A_t = \{x : U_t(x) \geq \ell_t\}$
4:     $x_{t+1} \leftarrow \arg\max_{x \in A_t}[U_t(x) - L \cdot \min_i d(x, x_i)]$
5:     Query $x_{t+1}$, observe $y_{t+1}$, update statistics
6:     Replicate active points with $r_i(t) > \beta_{\text{sched}}(t)$
7: **end for**
8: **Output:** $\hat{x}_T = \arg\max_i \hat{\mu}_i(T)$, certificate $A_T$

---

we use CMA-ES with 10 random restarts within $A_t$ (see Appendix B for details). For $d \leq 3$, we additionally use Delaunay triangulation to identify candidate optima at Voronoi vertices. Membership in $A_t$ is exact: checking $U_t(x) \geq \ell_t$ requires $O(N_t)$ time. Approximate maximization may slow convergence of $\eta_t$ but does not invalidate certificates: any $x \notin A_t$ remains certifiably suboptimal regardless of which $x \in A_t$ is queried.

**Replication Strategy.** When an active point $x_i$ has $r_i(t) > \beta_{\text{sched}}(t)$, we allocate $\lceil(r_i(t)/\beta_{\text{sched}}(t))^2\rceil$ additional samples to reduce its confidence radius. This ensures all active points have comparable confidence, preventing any single point from dominating the envelope.

## 4. Theoretical Analysis

Having described the algorithm, we now establish its theoretical guarantees. We show that the active set is contained in the near-optimal set, its volume shrinks at a controlled rate, and this yields instance-dependent sample complexity. All results hold on the good event $\mathcal{E}$ where $|f(x_i) - \hat{\mu}_i(t)| \leq r_i(t)$ for all $t, i$. All proofs are deferred to Appendix C.

**Lemma 4.1** (Good event). *With* $r_i(t) = \sigma\sqrt{2\log(2N_tT/\delta)/n_i}$, *we have* $\mathbb{P}[\mathcal{E}] \geq 1 - \delta$.

**Lemma 4.2** (UCB envelope is valid). *On* $\mathcal{E}$, *for all* $x \in \mathcal{X}$: $f(x) \leq U_t(x)$.

*Proof sketch.* For any sampled $x_i$, on $\mathcal{E}$: $f(x_i) \leq \text{UCB}_i(t)$. By Lipschitz continuity: $f(x) \leq f(x_i) + L \cdot d(x, x_i) \leq \text{UCB}_i(t) + L \cdot d(x, x_i)$. Taking min over $i$ gives $f(x) \leq U_t(x)$. $\qquad\square$

**Lemma 4.3** (Envelope slack bound). *On* $\mathcal{E}$, *for all* $x \in \mathcal{X}$: $U_t(x) \leq f(x) + 2\rho_t(x)$, *where* $\rho_t(x) = \min_i\{r_i(t) + L \cdot d(x, x_i)\}$.

*Remark* 4.4 (Slack in envelope bound). The factor of 2 arises from applying Lipschitz continuity ($f(x_i) \leq f(x) +$

$L \cdot d(x, x_i)$) after bounding $\mathrm{UCB}_i \leq f(x_i) + 2r_i$. A tighter bound separating the confidence and distance terms is possible but complicates notation without affecting rate dependencies. Constants throughout are not optimized.

**Theorem 4.5** (Active set containment). *On $\mathcal{E}$, $A_t \subseteq \{x : f(x) \geq f^* - 2\Delta_t\}$ where $\Delta_t = \sup_{x \in A_t} \rho_t(x) + (f^* - \ell_t)$.*

*Proof sketch.* On $\mathcal{E}$, $\ell_t = \max_i \mathrm{LCB}_i(t) \leq f^*$ since each $\mathrm{LCB}_i(t) \leq f(x_i) \leq f^*$. For $x \in A_t$, by definition $U_t(x) \geq \ell_t$. Applying Lemma 4.3: $f(x) + 2\rho_t(x) \geq U_t(x) \geq \ell_t$. Rearranging: $f(x) \geq \ell_t - 2\rho_t(x) = f^* - (f^* - \ell_t) - 2\rho_t(x) \geq f^* - 2\Delta_t$. $\qquad\square$

The containment theorem bounds how far active points can be from optimal. To translate this into a volume bound, we introduce two key quantities: the covering radius $\eta_t = \sup_{x \in A_t} \min_i d(x, x_i)$ measuring how well samples cover $A_t$, and the active confidence radius $\beta_t = \max_{i : x_i \text{ active}} r_i(t)$ measuring confidence precision.

**Theorem 4.6** (Shrinkage theorem). *Under Assumptions 2.1–2.4, on $\mathcal{E}$:*

$$\mathrm{Vol}(A_t) \leq C \cdot \big(2(\beta_t + L\eta_t) + \gamma_t\big)^{d-\alpha}, \qquad (3)$$

*where $\gamma_t = f^* - \ell_t$.*

*Proof sketch.* From Theorem 4.5, $A_t \subseteq \{x : f(x) \geq f^* - 2\Delta_t\}$. For $x \in A_t$, $\rho_t(x) \leq \beta_t + L\eta_t$ (the worst-case slack from active-point confidence plus covering distance), so $\Delta_t \leq \beta_t + L\eta_t + \gamma_t/2$. Applying Assumption 2.4 with $\varepsilon = 2\Delta_t$: $\mathrm{Vol}(A_t) \leq C(2\Delta_t)^{d-\alpha} \leq C(2(\beta_t + L\eta_t) + \gamma_t)^{d-\alpha}$. $\qquad\square$

This makes pruning measurable: $\beta_t$ is controlled by replication, $\eta_t$ by the query rule, $\gamma_t$ by best-point improvement. All three quantities can be computed during the run, enabling practitioners to monitor progress. We henceforth define the *computable gap proxy*

$$\varepsilon_t := 2(\beta_t + L\eta_t) + \gamma_t, \qquad (4)$$

which serves as our certified anytime stopping signal.

*Remark* 4.7 (Certificate validity vs. progress estimation). The certificate itself is the set membership rule $x \in A_t \Leftrightarrow U_t(x) \geq \ell_t$, which is exact given $(\hat{\mu}_i, r_i)$ and does not depend on any volume estimator. Approximations (grid/Monte Carlo) are used only to estimate $\mathrm{Vol}(A_t)$ for monitoring; certificate validity is unaffected by volume estimation errors. The primary anytime certified stopping rule uses $\varepsilon_t$ from (4).

### 4.1. Algorithmic Guarantee under Approximate Optimization

The shrinkage theorem gives a bound on $\mathrm{Vol}(A_t)$ *conditional on* the values of $\beta_t, \eta_t, \gamma_t$ reached. We now bridge this to Algorithm 1's actual mechanics by formalizing the approximation regime in which the inner score maximization operates, and showing that the algorithm drives the three quantities to the required scales.

**Definition 4.8** ($C$-approximate oracle). An optimizer is a $C$-approximate oracle (with $C \geq 1$) if it returns $\tilde{x}_{t+1} \in A_t$ satisfying

$$\mathrm{score}(\tilde{x}_{t+1}) \geq \frac{1}{C} \sup_{x \in A_t} \mathrm{score}(x).$$

**Lemma 4.9** (Algorithm 1 drives $\beta_t, \eta_t, \gamma_t$). *Under Assumptions 2.1–2.4 and a $C$-approximate oracle:*

*(i)* **Confidence.** *The replication rule guarantees $\beta_t \leq \beta_{\mathrm{sched}}(t) = \sigma\sqrt{2\log(2T^2/\delta)/t}$ after $n_i \geq \lceil (r_i/\beta_{\mathrm{sched}})^2 \rceil$ extra samples at each active point, yielding $\beta_t = O\big(\sigma\sqrt{\log(T/\delta)/t}\big)$.*

*(ii)* **Coverage.** *The coverage penalty $-L \cdot \min_i d(x, x_i)$ in $\mathrm{score}(x)$ ensures any uncovered ball of radius $\eta$ in $A_t$ attains $\mathrm{score} \geq U_t^{\min} + L\eta$. After $N_t = O((D/\eta)^{d-\alpha})$ distinct active locations (using $\mathrm{Vol}(A_t) \leq C\varepsilon_t^{d-\alpha}$), the $C$-approximate oracle drives $\eta_t \leq C\eta$. Approximation enters only through constants, not exponents.*

*(iii)* **Gap.** *Since $\ell_t = \max_i \mathrm{LCB}_i(t)$ is non-decreasing in $t$, and replication concentrates additional samples at the empirical maximizer, $\gamma_t = f^* - \ell_t \to 0$ at rate $O(\beta_{\mathrm{sched}}(t))$ once $\eta_t$ has reached the support of $x^*$.*

**Theorem 4.10** (Sample complexity, refined). *Under Assumptions 2.1–2.4 and a $C$-approximate oracle, CGP achieves $r_T \leq \varepsilon$ with probability $\geq 1 - \delta$ using*

$$T = \tilde{O}\big(C^d L^d \varepsilon^{-(2+\alpha)} \log(1/\delta)\big).$$

*When $\alpha < d$, this improves upon the worst case $\tilde{O}(\varepsilon^{-(2+d)})$ rate. The approximation factor $C$ enters only through a $C^d$ constant, not the exponent in $\varepsilon$.*

The following lower bound shows that our sample complexity is optimal up to logarithmic factors. Crucially, the construction exhibits near-optimality dimension *exactly $\alpha$*, so the $\alpha$-dependence is intrinsic rather than imposed.

**Theorem 4.11** (Lower bound, $\alpha$-tight). *Fix $\alpha \in (0, d]$, $L \geq 1$. There is a family $\{f_{i^*}\}_{i^* \in [M]}$ with $M = \lceil \varepsilon^{-\alpha} \rceil$ and Lipschitz constant $L$ such that (i) every $f_{i^*}$ has near-optimality dimension exactly $\alpha$, and (ii) any algorithm achieving $r_T \leq \varepsilon$ with probability $\geq 2/3$ requires*

$$T = \Omega\big(\varepsilon^{-(2+\alpha)}\big).$$

## 4.2. Volume-Based Stopping under Lower Regularity

A natural question is whether $\mathrm{Vol}(A_t)$ alone can certify $\varepsilon$-optimality. The shrinkage theorem bounds $\mathrm{Vol}(A_t)$ *above* in terms of $\varepsilon_t$, but does not invert this: small volume need not imply small regret without additional assumptions. We resolve this with a complementary lower-regularity condition.

**Assumption 4.12** (Lower regularity). There exist $c' > 0$ and the same $\alpha \in [0, d]$ as in Assumption 2.4 such that for all $\varepsilon > 0$:

$$\mathrm{Vol}(\{x \in \mathcal{X} : f(x) \geq f^* - \varepsilon\}) \geq c' \, \varepsilon^{d-\alpha}.$$

This is mild: near a nondegenerate maximizer with $\alpha = d/2$, near-optimal volume scales as $\Theta(\varepsilon^{d/2})$, satisfying both Assumptions 2.4 and 4.12.

**Theorem 4.13** (Volume-based stopping). *Under Assumptions 2.1–2.4 and Assumption 4.12, on $\mathcal{E}$:*

$$r_T \leq \left(\frac{\mathrm{Vol}(A_T)}{c'}\right)^{1/(d-\alpha)} + \frac{\gamma_T}{2}.$$

*Hence stopping when $\mathrm{Vol}(A_T) \leq c'(\varepsilon/2)^{d-\alpha}$ and $\gamma_T \leq \varepsilon$ guarantees $r_T \leq \varepsilon$.*

*Remark* 4.14 (Scope of volume-based stopping). Without Assumption 4.12, $\mathrm{Vol}(A_t)$ is a correlated diagnostic but not a certified stopping signal. With Assumption 4.12, volume-based stopping is rigorously justified. In practice we report both: $\varepsilon_t$ is always-valid; $\mathrm{Vol}(A_t)$ becomes a valid stopping rule under the additional lower-regularity condition that holds at nondegenerate maxima.

This establishes CGP is minimax optimal up to logarithmic factors. A key property is anytime validity: at any $t$, any $x \notin A_t$ satisfies $f(x) < f^* - \varepsilon_t$ for computable $\varepsilon_t > 0$. Full proofs are in Appendix C.

## 5. CGP-Adaptive: Learning $L$ Online

The theoretical results above assume known $L$, which is often unavailable in practice. Underestimating $L$ invalidates certificates, while overestimating is safe but conservative. We develop CGP-Adaptive (Algorithm 2), which learns $L$ online via a doubling scheme with provable guarantees.

The key insight is that Lipschitz violations are detectable. If $|\hat{\mu}_i - \hat{\mu}_j| - 2(r_i + r_j) > \hat{L} \cdot d(x_i, x_j)$, then $\hat{L}$ underestimates $L$ with high probability. CGP-Adaptive uses a doubling scheme: start with conservative $\hat{L}_0$, and upon detecting a violation, double $\hat{L}$.

**Theorem 5.1** (Adaptive $L$ guarantee: learning regime). *Let $L^* = \sup_{x \neq y} |f(x) - f(y)|/d(x, y)$ be the true Lipschitz constant. CGP-Adaptive with initial $\hat{L}_0 \leq L^*$ (learning from underestimation) satisfies:*

---

**Algorithm 2** CGP-Adaptive

**Require:** Domain $\mathcal{X}$, initial estimate $\hat{L}_0$, noise $\sigma$, budget $T$, confidence $\delta$
1: $\hat{L} \leftarrow \hat{L}_0$, $k \leftarrow 0$ (doubling counter)
2: **for** $t = 1, \ldots, T$ **do**
3:     Run CGP iteration with current $\hat{L}$
4:     **for** all pairs $(i, j)$ with $n_i, n_j \geq \log(T/\delta)$ **do**
5:         **if** $|\hat{\mu}_i - \hat{\mu}_j| - 2(r_i + r_j) > \hat{L} \cdot d(x_i, x_j)$ **then**
6:             $\hat{L} \leftarrow 2\hat{L}$, $k \leftarrow k + 1$ {Doubling event}
7:             Recompute $A_t$ with new $\hat{L}$
8:         **end if**
9:     **end for**
10: **end for**

---

1. *The number of doubling events is at most $K = \lceil \log_2(L^*/\hat{L}_0) \rceil$.*

2. *After all doublings, $\hat{L} \in [L^*, 2L^*]$ with probability $\geq 1 - \delta$.*

3. *The total sample complexity is $T = \tilde{O}(\varepsilon^{-(2+\alpha)} \cdot K)$, i.e., $O(\log(L^*/\hat{L}_0))$ multiplicative overhead.*

4. **Certificate validity:** *Certificates are valid only after the final doubling (when $\hat{L} \geq L^*$). Before this, certificates may falsely exclude near-optimal points.*

*Remark* 5.2 (Anytime-valid certificates). The anytime-valid certificate property in our abstract and introduction refers to **vanilla CGP** with known or conservatively bounded $L$ (i.e., $\hat{L}_0 \geq L^*$). For CGP-Adaptive with $\hat{L}_0 < L^*$, certificates become valid only after the final doubling event (Theorem 5.1(4)). In practice, initializing $\hat{L}_0$ from finite differences on 10 Sobol samples (scaled by $\sqrt{2}$) yields $\hat{L}_0 \geq L^*$ in 94% of our benchmarks, providing anytime validity from iteration 1. CGP-Adaptive exposes a binary certificates_valid flag that transitions permanently from False to True after the final doubling, allowing users to gate downstream decisions accordingly.

This is the first provably correct adaptive $L$ estimation for Lipschitz optimization with certificates. Prior work (Malherbe & Vayatis, 2017) estimates $L$ but without guarantees on certificate validity. Table 5 shows CGP-Adaptive matches oracle performance (known $L$) within 8% while being robust to $100\times$ underestimation of initial $\hat{L}_0$.

**Adaptive noise estimation.** The same doubling principle extends to unknown $\sigma$: replicated observations at active points (already collected by the replication rule) yield a sample variance $\hat{\sigma}^2$ that concentrates at rate $O(1/\sqrt{n_i})$. Replacing $\sigma$ with an upper confidence bound on $\hat{\sigma}$ preserves certificate validity at $O(\log T)$ overhead. On Hartmann-6 with $\sigma$ estimated online, regret degrades only $\sim 7\%$ vs. the known-$\sigma$ baseline; see Appendix D.

*Table 3.* Simple regret ($\times 10^{-2}$) at $T = 200$. Bold: best; $^\dagger$: significant vs second-best.

| Method | Needle | Branin | Hartmann | Ackley | Levy | Rosen. | SVM |
|--------|--------|--------|----------|--------|------|--------|-----|
| Random | 8.2 | 12.1 | 15.3 | 22.4 | 18.7 | 14.2 | 11.2 |
| GP-UCB | 2.1 | 1.8 | 4.2 | 12.3 | 5.1 | 4.8 | 3.9 |
| TuRBO | 1.8 | 2.1 | 3.1 | 9.8 | 4.3 | 3.9 | 3.2 |
| HEBO | 1.9 | 1.6 | 3.3 | 9.4 | 4.1 | 3.7 | 3.1 |
| BORE | 2.0 | 1.9 | 3.5 | 10.1 | 4.5 | 4.1 | 3.4 |
| HOO | 3.4 | 5.2 | 8.7 | 14.2 | 9.8 | 8.1 | 7.1 |
| CGP | 1.2 | 2.0 | 2.9 | 8.1 | 3.8 | 3.8 | 2.8 |
| CGP-A | 1.3 | 2.1 | 3.0 | 8.3 | 3.9 | 3.9 | 2.9 |
| **CGP-H** | **1.1**$^\dagger$ | **1.4**$^\dagger$ | **2.7**$^\dagger$ | **7.8**$^\dagger$ | **3.5**$^\dagger$ | **3.4**$^\dagger$ | **2.6**$^\dagger$ |

# 6. CGP-TR: Trust Regions for High Dimensions

CGP-Adaptive addresses the unknown $L$ problem, but another challenge remains: scalability. The covering number of $A_t$ grows as $O(\eta^{-d})$, making CGP intractable for $d > 15$. To enable high-dimensional optimization, we develop CGP-TR (Algorithm 3), which maintains local certificates within trust regions that adapt based on observed progress.

The key insight is that certificates need not be global. A local certificate $A_t^{\mathcal{T}}$ within trust region $\mathcal{T} \subset \mathcal{X}$ still provides valid bounds for $\arg\max_{x \in \mathcal{T}} f(x)$. CGP-TR maintains multiple trust regions $\{\mathcal{T}_1, \ldots, \mathcal{T}_m\}$ centered at promising points, with radii that expand on success and contract on failure (following TuRBO (Eriksson et al., 2019)).

**Certified restarts.** We restart a trust region only when it is *certifiably suboptimal*: if $u_t^{(j)} := \max_{x \in \mathcal{T}_j} U_t(x)$ satisfies $u_t^{(j)} < \ell_t$ where $\ell_t := \max_i \text{LCB}_i(t)$, then with high probability $\sup_{x \in \mathcal{T}_j} f(x) < f(x^*)$, so $\mathcal{T}_j$ cannot contain $x^*$ and can be safely restarted. This certified restart rule ensures that regions containing $x^*$ are never falsely eliminated. In our implementation, contraction is lower-bounded by $r_{\min}$ and centers are fixed, so a region that contains $x^*$ cannot be contracted to exclude it; restarts occur only via the certified condition $u_t^{(j)} < \ell_t$.

**Theorem 6.1** (CGP-TR with certified restarts: correctness and allocation). *Assume the good event $\mathcal{E}$ holds for the confidence bounds used to construct $U_t$ and $\ell_t$. CGP-TR uses certified restarts: restart $\mathcal{T}_j$ only if $u_t^{(j)} := \max_{x \in \mathcal{T}_j} U_t(x) < \ell_t$.*

*Let $\mathcal{T}^*$ be a trust region that contains $x^*$ at some time and is not contracted to exclude $x^*$ (e.g., contraction is lower-bounded by $r_{\min}$ and the center remains fixed). Then:*

1. *(No false restarts) On $\mathcal{E}$, $\mathcal{T}^*$ is never restarted by the certified rule.*

2. *(Local certificate) Conditioned on receiving $T^*$ eval-*

---

**Algorithm 3** CGP-TR (Trust Region with Certified Restarts)

**Require:** Domain $\mathcal{X}$, $L$, $\sigma$, budget $T$, initial radius $r_0$, $n_{\text{trust}}$ regions

1: Initialize $n_{\text{trust}}$ trust regions at Sobol points with radius $r_0$
2: **for** $t = 1, \ldots, T$ **do**
3:     Compute $\ell_t := \max_{i \leq N_t} \text{LCB}_i(t)$ (global lower certificate)
4:     Select trust region $\mathcal{T}_j$ with highest $u_t^{(j)} := \max_{x \in \mathcal{T}_j} U_t(x)$
5:     Run CGP within $\mathcal{T}_j$: compute local $A_t^{(j)} = \{x \in \mathcal{T}_j : U_t(x) \geq \ell_t^{(j)}\}$
6:     Query $x_{t+1} \in A_t^{(j)}$, observe $y_{t+1}$
7:     **if** $u_t^{(j)} < \ell_t$ **then**
8:         **Certified restart:** restart $\mathcal{T}_j$ at a new Sobol point with radius $r_0$
9:     **else if** improvement in $\mathcal{T}_j$ **then**
10:       Expand: $r_j \leftarrow \min(2r_j, D/2)$
11:     **else if** no improvement for $\tau_{\text{fail}}$ iterations **then**
12:       Contract: $r_j \leftarrow \max(r_j/2, r_{\min})$
13:     **end if**
14: **end for**
15: **Output:** Best point across all regions, local certificate $A_T^{(j^*)}$

---

*uations inside $\mathcal{T}^*$, the local active set $A_t^{(\mathcal{T}^*)} = \{x \in \mathcal{T}^* : U_t(x) \geq \ell_t^{(\mathcal{T}^*)}\}$ satisfies the same containment/shrinkage/sample-complexity bounds as CGP on the restricted domain $\mathcal{T}^*$.*

3. *(Allocation bound) Define the region gap $\Delta_j := f^* - \sup_{x \in \mathcal{T}_j} f(x)$ (with $\Delta_j > 0$ for suboptimal regions). Assume each region runs CGP with replication ensuring its maximal active-point confidence radius after $n$ within-region samples satisfies $\beta_j(n) \leq c_\sigma \sqrt{\log(c_T/\delta)/n}$ for constants $c_\sigma, c_T$ matching the paper's confidence schedule. If the region radii are eventually bounded so that $L \cdot \text{diam}(\mathcal{T}_j) \leq \Delta_j/8$,*

*then any suboptimal region $j$ is selected at most*

$$N_j \leq \frac{64 c_\sigma^2}{\Delta_j^2} \log\left(\frac{c_T}{\delta}\right) + 1$$

*times before it is eliminated by the certified restart rule.*

The key advantage is that covering $\mathcal{T}_j$ requires $O((r_j/\eta)^d)$ points, and since $r_j \ll D$, this is tractable even for large $d$. With $n_{\text{trust}} = O(\log T)$ regions, CGP-TR explores globally while maintaining local certificates. The allocation bound (Theorem 6.1, item 3) ensures that suboptimal regions receive only $O(\log T/\Delta_j^2)$ evaluations before certified elimination, preventing wasted samples.

CGP-TR provides local rather than global certificates, but the certified restart rule guarantees that the region containing $x^*$ is never falsely eliminated. This enables scaling to $d = 50$ to $100$ where global Lipschitz methods fail entirely.

## 7. CGP-Hybrid: Best of Both Worlds

While CGP-TR addresses scalability, some functions exhibit local smoothness that GPs can exploit more effectively than Lipschitz methods. CGP-Hybrid (Algorithm 4) preserves CGP's anytime certificates while allowing any optimizer to refine within the certified active set. The key point is modularity: Phase 1 constructs a certificate $A_t$; Phase 2 performs additional optimization restricted to $A_t$ without affecting certificate validity. We instantiate Phase 2 with GP-UCB when local smoothness is detected, but other optimizers can be used. This design captures the best of both worlds: CGP's explicit pruning guarantees and GP's ability to exploit local smoothness when present.

Define the *effective smoothness ratio* $\rho_t = \hat{L}_{\text{local}}(t)/\hat{L}_{\text{global}}$, where $\hat{L}_{\text{local}}(t)$ is estimated from points within $A_t$. When $\rho_t < 0.5$, the function is significantly smoother near the optimum, and GP refinement is beneficial.

**Proposition 7.1** (Hybrid guarantee). *CGP-Hybrid achieves:*

1. *If $\rho \geq 0.5$: same guarantee as CGP, $T = \tilde{O}(\varepsilon^{-(2+\alpha)})$.*

2. *If $\rho < 0.5$: after CGP reduces $A_t$ to volume $V$, GP-UCB operates within a restricted domain of effective diameter $O(V^{1/d})$. The additional sample complexity depends on the GP kernel's information gain $\gamma_T$ over $A_t$; empirically, this yields faster convergence than continuing CGP when the function is locally smooth.*

3. *The certificate $A_T$ from Phase 1 remains valid regardless of Phase 2 method.*

**Proposition 7.2** (Certificate invariance under Phase 2). *The certificate $A_t$ computed by CGP in Phase 1 remains valid regardless of the Phase-2 optimizer, since validity depends*

---

**Algorithm 4** CGP-Hybrid

---

**Require:** Domain $\mathcal{X}$, $L$, $\sigma$, budget $T$, switch threshold $\rho_{\text{thresh}} = 0.5$
1: Phase 1: Run CGP until $\text{Vol}(A_t) < 0.1 \cdot \text{Vol}(\mathcal{X})$ or $t > T/3$
2: Estimate $\rho_t = \hat{L}_{\text{local}}(t)/\hat{L}_{\text{global}}$
3: **if** $\rho_t < \rho_{\text{thresh}}$ **then**
4:     Phase 2: Switch to GP-UCB within $A_t$ (GP refinement)
5:     Fit GP to points in $A_t$, continue with GP-UCB acquisition
6: **else**
7:     Phase 2: Continue CGP within $A_t$
8: **end if**
9: **Output:** Best point, certificate $A_T$ (from CGP phase)

---

*only on the confidence bounds and Lipschitz envelope used to define $U_t$ and $\ell_t$. Specifically, any point $x \notin A_t$ satisfies $f(x) < f^* - \varepsilon_t$ with high probability, where $\varepsilon_t$ is computable from Phase 1 quantities alone.*

**Robustness to the smoothness heuristic.** The threshold $\rho_{\text{thresh}} = 0.5$ is chosen by cross-validation. A potential concern is a function that is globally smooth but has a sharp narrow peak at $x^*$: if Phase 1 has not yet localized $A_t$ near $x^*$, $\rho_t$ may misleadingly look small. We verify this scenario empirically (Appendix E) on $f(x) = -\|x\| + 0.5 \cdot \mathbf{1}[\|x\| < 0.05]$, where Phase 1's concentration ensures $\rho_t$ is estimated within $A_t$ (which contains $x^*$), and the heuristic stays in CGP (regret 1.4 vs. GP-UCB's 4.2). Importantly, even if the switch misfires, the certificate $A_T$ from Phase 1 remains valid (Proposition 7.2); only convergence speed is affected. Threshold sensitivity is mild: varying $\rho_{\text{thresh}} \in [0.3, 0.7]$ changes regret by $< 5\%$ on borderline cases (Appendix E).

The key insight is that CGP's certificate remains valid even when switching to GP: $A_t$ still contains $x^*$ with high probability, so GP refinement within $A_t$ is safe. This provides the best of both worlds: CGP's certificates and pruning efficiency when $\rho \geq 0.5$, and GP's smoothness exploitation when $\rho < 0.5$.

Table 6 shows CGP-Hybrid wins on all 12 benchmarks, including Branin ($\rho = 0.31$) and Rosenbrock ($\rho = 0.28$) where it detects low $\rho$ and switches to GP refinement, and Needle ($\rho = 0.98$) where it stays with CGP.

## 8. Related Work

Our work is primarily grounded in the literature on Lipschitz bandits and global optimization. Foundational approaches, such as the continuum-armed bandits of Kleinberg et al. (2008) and the X-armed bandit framework of Bubeck et al. (2011), utilize zooming mechanisms to achieve re-

Table 4. High-dimensional benchmarks ($d > 20$) at $T = 500$.

| Method | Rover-60 | NAS-36 | Ant-100 |
|---|---|---|---|
| Random | $42.1 \pm 1.2$ | $38.4 \pm 0.9$ | $51.2 \pm 1.4$ |
| TuRBO | $12.4 \pm 0.4$ | $11.2 \pm 0.3$ | $18.7 \pm 0.6$ |
| HEBO | $14.1 \pm 0.5$ | $12.8 \pm 0.4$ | $21.3 \pm 0.7$ |
| CMA-ES | $15.8 \pm 0.6$ | $14.1 \pm 0.5$ | $19.4 \pm 0.6$ |
| CGP | (intractable for $d > 15$) | | |
| CGP-TR | $\mathbf{11.2 \pm 0.3}^{\dagger}$ | $\mathbf{10.4 \pm 0.3}^{\dagger}$ | $\mathbf{17.1 \pm 0.5}^{\dagger}$ |

† CGP-TR additionally provides local optimality certificates.

gret bounds depending on the near-optimality dimension. These concepts were refined for deterministic and stochastic settings via tree-based algorithms like DOO/SOO (Munos, 2011) and StoSOO (Valko et al., 2013). However, a key distinction is that zooming algorithms maintain pruning implicitly as an analysis artifact, whereas CGP exposes the active set $A_t$ as a computable object with measurable volume. This explicit geometric approach also relates to partition-based global optimization methods like DIRECT (Jones et al., 1993) and LIPO (Malherbe & Vayatis, 2017). While LIPO addresses the unknown Lipschitz constant, it lacks the certificate guarantees provided by our CGP-Adaptive doubling scheme (Shihab et al., 2025c). Our theoretical analysis further draws on sample complexity results under margin conditions from the finite-arm setting (Audibert & Bubeck, 2010; Jamieson & Nowak, 2014; Shihab et al., 2025b), adapting confidence bounds from the UCB framework (Auer et al., 2002) to continuous spaces with explicit uncertainty representation similar to safe optimization level-sets (Sui et al., 2015).

**Comparison with GP-based methods.** GP-UCB (Srinivas et al., 2010) achieves cumulative regret $\tilde{O}(\sqrt{T\gamma_T})$ where $\gamma_T$ is the kernel's maximum information gain. For Matérn-$\nu$ with $\nu = 1/2$ (the kernel matching Lipschitz functions), $\gamma_T = O(T^{(d+1)/(d+2)})$, yielding simple regret $\tilde{O}(T^{-1/(d+2)})$, equivalently $T = \tilde{O}(\varepsilon^{-(d+2)})$. This is strictly worse than CGP's $\tilde{O}(\varepsilon^{-(2+\alpha)})$ whenever $\alpha < d$—the typical case for isolated maxima ($\alpha = d/2$). For Hartmann-6 (empirical $\hat{\alpha} \approx 2.4 < 6$), CGP requires $\tilde{O}(\varepsilon^{-4.4})$ vs. GP-UCB's $\tilde{O}(\varepsilon^{-8})$, consistent with the empirical gap in Table 3. GP-UCB also requires kernel specification, which implicitly enforces smoothness beyond Lipschitz continuity. Conversely, when $f$ has higher-order smoothness, GP-UCB exploits it through the kernel while CGP cannot—motivating the hybrid in Section 7.

In the broader context of black-box optimization, Bayesian methods provide the standard alternative for uncertainty quantification. Classic approaches like GP-UCB (Srinivas et al., 2010), Entropy Search (Hennig & Schuler, 2012; Hernández-Lobato et al., 2014), and Thompson Sampling (Thompson, 1933) offer strong performance but scale cubically with observations. To address high-dimensional scaling, recent work has introduced trust regions (TuRBO) (Eriksson et al., 2019) and nonstationary priors (HEBO) (Cowen-Rivers et al., 2022). More recently, advanced heuristics such as Bounce (Papenmeier et al., 2024) have improved geometric adaptation for mixed spaces, while Prior-Fitted Networks (Müller et al., 2023) and generative diffusion models like Diff-BBO (Wu et al., 2024) exploit massive pretraining to minimize regret rapidly. While these emerging methods achieve impressive empirical results, they remain fundamentally heuristic, lacking the computable stopping criteria or active set containment guarantees that are central to CGP. We therefore focus our comparison on established baselines to isolate the specific utility of our certification mechanism, distinguishing our approach from heuristic hyperparameter tuners like Hyperband (Li et al., 2018) by prioritizing provable safety over raw speed.

## 9. Experiments

We evaluate CGP variants on 12 benchmarks spanning $d \in [2, 100]$, measuring simple regret, certificate utility, and scalability. Code and experiment scripts are available at the project repository linked in the supplementary material. We compare against 9 baselines: Random Search, GP-UCB, TuRBO, HEBO, BORE, HOO, StoSOO, LIPO, and SAASBO (see Appendix I for configurations; the full $9 \times 12$ comparison table appears in Appendix F). We evaluate on 12 benchmarks spanning $d \in [2, 100]$: low-dimensional (Needle-2D, Branin, Hartmann-6, Levy-5, Rosenbrock-4), medium-dimensional (Ackley-10, SVM-RBF-6, LunarLander-12), and high-dimensional (Rover-60, NAS-36, MuJoCo-Ant-100). All experiments use 30 runs with $\sigma = 0.1$ noise; we report mean $\pm$ SE with Bonferroni-corrected $t$-tests ($p < 0.05$).

Table 3 shows CGP-Hybrid performs best among tested methods on all 7 low and medium-dimensional benchmarks. On Branin ($\hat{\rho} = 0.31$) and Rosenbrock ($\hat{\rho} = 0.28$) where vanilla CGP lost to HEBO, CGP-Hybrid detects $\rho < 0.5$ and switches to GP refinement, achieving 12% and 8% improvement over HEBO respectively. On Needle ($\hat{\rho} \approx 0.98$), CGP-Hybrid stays with CGP and matches vanilla CGP performance. For high-dimensional problems, Table 4 demonstrates CGP-TR scales to $d = 100$ while outperforming TuRBO by 9 to 12%. Critically, CGP-TR provides local certificates within trust regions, enabling principled stopping—a capability TuRBO lacks. Regarding adaptive estimation, Table 5 shows CGP-Adaptive is robust to initial underestimation: even with $\hat{L}_0 = L^*/100$, performance degrades only 10% with 7 doublings, validating Theorem 5.1's $O(\log(L^*/\hat{L}_0))$ overhead. Finally, Table 6 confirms CGP-Hybrid correctly identifies when to switch: Branin and

*Table 5.* CGP-Adaptive with varying initial $\hat{L}_0$. Robust to $100\times$ underestimation.

| Initial $\hat{L}_0$ | Doublings | Final $\hat{L}/L^*$ | Regret ($\times 10^{-2}$) | Overhead |
|---|---|---|---|---|
| $L^*$ (oracle) | 0 | 1.0 | $2.9 \pm 0.1$ | $1.0\times$ |
| $L^*/2$ | 1 | 1.0 | $3.0 \pm 0.1$ | $1.03\times$ |
| $L^*/10$ | 4 | 1.6 | $3.1 \pm 0.1$ | $1.07\times$ |
| $L^*/100$ | 7 | 1.3 | $3.2 \pm 0.2$ | $1.12\times$ |
| LIPO (adaptive) | – | – | $6.2 \pm 0.3$ | – |

*Table 6.* CGP-Hybrid smoothness detection. $\rho < 0.5$ triggers GP refinement.

| Benchmark | $\hat{\rho}$ | Phase 2 | CGP-H Regret | Best Baseline |
|---|---|---|---|---|
| Needle-2D | 0.98 | CGP | $1.1 \pm 0.1$ | 1.8 (TuRBO) |
| Branin | 0.31 | GP | $1.4 \pm 0.1$ | 1.6 (HEBO) |
| Hartmann-6 | 0.72 | CGP | $2.7 \pm 0.1$ | 3.1 (TuRBO) |
| Rosenbrock | 0.28 | GP | $3.4 \pm 0.1$ | 3.7 (HEBO) |
| Ackley-10 | 0.85 | CGP | $7.8 \pm 0.3$ | 9.4 (HEBO) |
| Levy-5 | 0.67 | CGP | $3.5 \pm 0.2$ | 4.1 (HEBO) |
| SVM-RBF | 0.74 | CGP | $2.6 \pm 0.1$ | 3.1 (HEBO) |
| LunarLander | 0.81 | CGP | $6.1 \pm 0.3$ | 7.0 (HEBO) |

Rosenbrock trigger GP refinement, achieving 12% and 8% improvement over HEBO. Benchmarks with $\rho > 0.5$ stay with CGP, maintaining certificate validity.

**Shrinkage validation.** Across all benchmarks, we observe $\text{Vol}(A_t)$ shrinks to $< 5\%$ by $T = 100$, with empirical decay rates closely matching the theoretical bound from Theorem 4.6. This confirms our analysis is tight and the margin condition captures the true problem difficulty. Table 9 demonstrates certificate utility: the certified $\varepsilon_t$-based rule (Theorem 4.10) saves 33% with no regret penalty, while the volume-based heuristic (valid under Assumption 4.12; Theorem 4.13) saves 59% at a controlled regret increase. No baseline provides such principled stopping rules. In $d > 20$, we use $\varepsilon_t$ as the primary criterion.

The nested-set ratio estimator (Appendix B.2) used for $d > 5$ has modest cost: 0.05s ($d = 6$), 0.15s ($d = 36$), 0.3s ($d = 60$), 0.6s ($d = 100$) per estimate. We invoke it once per 10 iterations as a monitoring signal, contributing $< 1\%$ of wall-clock time even at $d = 100$. Critically, certificate validity (Remark 4.7) does not depend on volume estimation: the set membership rule $U_t(x) \geq \ell_t$ is exact at $O(N_t)$ cost.

Beyond sample efficiency, CGP also offers computational advantages: it is $6$–$8\times$ faster than GP-based methods due to its $O(n)$ per-iteration cost versus GP's $O(n^3)$ (see Appendix H, Table 10). Finally, we ablate CGP's components to understand their individual contributions. Table 8 shows all components contribute: removing pruning certificates increases regret 78%, coverage penalty 44%, replication 33%. The "− GP refinement" row reflects the aggregate

contribution across the four hybrid-triggering benchmarks (Branin, Rosenbrock, and two others; on Hartmann-6 itself the hybrid stays in CGP). Appendix H (Table 11) further validates that $\hat{\alpha} < d$ across all benchmarks, confirming the margin condition holds and our complexity bounds apply.

## 10. Conclusion

We introduced Certificate-Guided Pruning (CGP), an algorithm for stochastic Lipschitz optimization that maintains explicit active sets with provable shrinkage guarantees. Under a margin condition with near-optimality dimension $\alpha$, we prove $\text{Vol}(A_t) \leq C \cdot (2(\beta_t + L\eta_t) + \gamma_t)^{d-\alpha}$, yielding sample complexity $\tilde{O}(\varepsilon^{-(2+\alpha)})$ together with anytime-valid certificates (for vanilla CGP with known or conservatively bounded $L$) and a certified stopping signal via the computable gap proxy $\varepsilon_t$. Three extensions broaden applicability: CGP-Adaptive learns $L$ online with $O(\log T)$ overhead (certificates becoming valid after the final doubling event), CGP-TR scales to $d > 50$ via trust regions with certified restarts, and CGP-Hybrid switches to GP refinement when local smoothness is detected while preserving Phase-1 certificates. The margin condition holds broadly: $\alpha = d/2$ for isolated maxima with nondegenerate Hessian, $\alpha = d/p$ for polynomial decay $\|x - x^*\|^p$, and can be estimated online from shrinkage trajectories. **Limitations** include requiring Lipschitz continuity and dimension constraints ($d \leq 15$ for vanilla CGP, $d \leq 100$ for CGP-TR); practical guidance is in Appendix G. Future directions include safe optimization using $A_t$ for safety certificates and CGP-TR with random embeddings for global high-dimensional certificates.

## Impact Statement

This paper introduces Certificate-Guided Pruning (CGP), a method designed to improve the sample efficiency of black-box optimization in resource-constrained settings. By providing explicit optimality certificates and principled stopping criteria, our approach significantly reduces the computational budget required for expensive tasks such as neural architecture search and simulation-based engineering, directly contributing to lower energy consumption and carbon footprints. Furthermore, the ability to certify suboptimal regions enhances reliability in safety-critical applications like robotics. However, practitioners must ensure the validity of the Lipschitz assumption, as violations could lead to the incorrect pruning of optimal solutions.

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

## A. Extended Problem Formulation

We restate the assumptions from Section 2 for completeness. Let $(\mathcal{X}, d)$ be a compact metric space with diameter $D = \sup_{x,y \in \mathcal{X}} d(x, y)$. We consider $\mathcal{X} = [0, 1]^d$ with Euclidean metric, though our results extend to general metric spaces. Let $f : \mathcal{X} \to [0, 1]$ be an unknown function satisfying:

**Assumption 2.1 (Lipschitz continuity).** There exists $L > 0$ such that for all $x, y \in \mathcal{X}$: $|f(x) - f(y)| \leq L \cdot d(x, y)$.

We observe $f$ through noisy queries: querying $x$ returns $y = f(x) + \epsilon$, where:

**Assumption 2.3 (Sub-Gaussian noise).** The noise $\epsilon$ is $\sigma$-sub-Gaussian: $\mathbb{E}[e^{\lambda \epsilon}] \leq e^{\lambda^2 \sigma^2 / 2}$ for all $\lambda \in \mathbb{R}$.

After $T$ samples, the algorithm outputs $\hat{x}_T \in \mathcal{X}$. The goal is to minimize simple regret $r_T = f(x^*) - f(\hat{x}_T)$, where $x^* \in \arg\max_{x \in \mathcal{X}} f(x)$. We seek PAC-style guarantees: with probability at least $1 - \delta$, achieve $r_T \leq \varepsilon$.

To obtain instance-dependent rates that improve upon worst case bounds, we assume margin structure:

**Assumption 2.4 (Margin / near-optimality dimension).** There exist $C > 0$ and $\alpha \in [0, d]$ such that for all $\varepsilon > 0$: $\mathrm{Vol}(\{x \in \mathcal{X} : f(x) \geq f^* - \varepsilon\}) \leq C \varepsilon^{d-\alpha}$.

The parameter $\alpha$ is the near-optimality dimension: smaller $\alpha$ corresponds to a sharper optimum (easier), while larger $\alpha$ corresponds to a broader near-optimal region (harder). The worst case is $\alpha = d$, which recovers the standard $d$-dimensional Lipschitz difficulty. This assumption is standard in bandit theory (Audibert & Bubeck, 2010; Bubeck et al., 2011; Lattimore & Szepesvári, 2020; Kaufmann et al., 2016; Garivier & Kaufmann, 2016).

**On differentiability.** Assumption 2.1 is strictly weaker than differentiability. Lipschitz functions can have kinks,

corners, and discontinuous gradients at every point. CGP's analysis relies only on the triangle inequality for Lipschitz propagation (Lemma 4.2) and the margin condition (Assumption 2.4); no derivatives appear. This is a structural advantage over GP-based methods that implicitly enforce higher-order smoothness via kernel choice. Two illustrative cases for $f(x) \approx f^* - c\|x - x^*\|^p$: $p = 1$ (a cone, non-differentiable at $x^*$, $\alpha = d$) and $p = 2$ (a paraboloid, smooth, $\alpha = d/2$). Both are handled by CGP under the same analysis, with the appropriate $\alpha$.

# B. Algorithm Implementation Details

## B.1. Score Maximization and Replication

CGP optimizes $\text{score}(x) = U_t(x) - \lambda \cdot \min_i d(x, x_i)$, which is piecewise linear. Since this is non-smooth, we use CMA-ES (Covariance Matrix Adaptation Evolution Strategy) with bounded domain and 10 random restarts within $A_t$. For $d \leq 3$, we additionally use Delaunay triangulation to identify candidate optima at Voronoi vertices.

Membership in $A_t$ is exact: given $\{x_i, \hat{\mu}_i, r_i\}$, checking $U_t(x) \geq \ell_t$ requires $O(N_t)$ time. Approximate maximization may slow convergence of $\eta_t$ but does not invalidate certificates: any $x \notin A_t$ remains certifiably suboptimal regardless of which $x \in A_t$ is queried.

In our experiments, CMA-ES typically achieves $C \leq 1.3$ within 10 restarts when measured against ground truth from dense grid search (in $d \leq 4$). Per Theorem 4.10, this contributes only a $C^d$ constant factor to the sample complexity bound, not the exponent.

**Replication Strategy.** When an active point $x_i$ has $r_i(t) > \beta_{\text{sched}}(t)$, we allocate $\lceil (r_i(t)/\beta_{\text{sched}}(t))^2 \rceil$ additional samples to reduce its confidence radius. This ensures all active points have comparable confidence, preventing any single point from dominating the envelope.

**KD-tree acceleration.** The envelope $U_t(x) = \min_i\{\text{UCB}_i(t) + L\,d(x, x_i)\}$ admits an $O(\log N_t)$ approximate evaluation via KD-trees: only nearby sampled points can be active in the minimum, so we prune candidates whose UCB plus minimum-possible distance exceeds the current best. This reduces effective per-iteration cost to $O(\log N_t)$ for large $N_t$, though the certificate membership check remains exact.

## B.2. Active Set Computation

For low dimensions ($d \leq 5$), we compute $A_t$ exactly using grid discretization with resolution $\eta = D/\sqrt[d]{N_{\text{grid}}}$ where $N_{\text{grid}} = 10^4$. For each grid point $x$, we evaluate $U_t(x)$ in $O(N_t)$ time and check if $U_t(x) \geq \ell_t$. The volume $\text{Vol}(A_t)$ is estimated as the fraction of grid points in $A_t$.

For higher dimensions ($d > 5$), uniform Monte Carlo becomes ineffective once $\text{Vol}(A_t)$ is small. We therefore estimate $\text{Vol}(A_t)$ via a nested-set ratio estimator (subset simulation): define thresholds $\ell_t - \tau_0 < \ell_t - \tau_1 < \cdots < \ell_t$ inducing nested sets

$$A_t^{(k)} = \{x : U_t(x) \geq \ell_t - \tau_k\}, \quad A_t^{(K)} = A_t.$$

We estimate

$$\text{Vol}(A_t) = \text{Vol}(A_t^{(0)}) \prod_{k=1}^{K} \mathbb{P}_{x \sim \text{Unif}(A_t^{(k-1)})}[x \in A_t^{(k)}],$$

sampling approximately uniformly from $A_t^{(k-1)}$ using a hit-and-run Markov chain with the membership oracle $U_t(x) \geq \ell_t - \tau_{k-1}$. This yields stable estimates even when $\text{Vol}(A_t)$ is very small. In all high-dimensional experiments, we report confidence intervals of $\log \text{Vol}(A_t)$ from repeated estimator runs. Concrete timings of this estimator are: 0.05s ($d = 6$, Hartmann), 0.15s ($d = 36$, NAS), 0.3s ($d = 60$, Rover), 0.6s ($d = 100$, Ant) per estimate. The estimator runs once per 10 iterations for monitoring.

Crucially, certificate validity (Remark 4.7) is independent of volume estimation accuracy: the set membership rule $x \in A_t \Leftrightarrow U_t(x) \geq \ell_t$ is exact.

**On volume-based stopping.** The shrinkage bound (Theorem 4.6) provides an upper bound on $\text{Vol}(A_t)$ as a function of the algorithmic gap proxy $\varepsilon_t := 2(\beta_t + L\eta_t) + \gamma_t$ and therefore supports using $\text{Vol}(A_t)$ as a practical *progress diagnostic*. Without additional assumptions, $\text{Vol}(A_t)$ alone does not yield an anytime upper bound on regret; under Assumption 4.12 (lower regularity), Theorem 4.13 provides a volume-based stopping rule with formal guarantee. In our experiments we therefore use $\varepsilon_t$ as the primary certificate-based stopping criterion and $\text{Vol}(A_t)$ as a secondary signal (whose validity is rigorous under nondegenerate-maximum geometry).

# C. Proofs

## C.1. Proof of Lemma 4.1

*Proof.* Define the good event $\mathcal{E} = \bigcap_{t=1}^{T} \bigcap_{i=1}^{N_t} \{|\hat{\mu}_i(t) - f(x_i)| \leq r_i(t)\}$.

*Step 1: Single-point concentration.* By Hoeffding's inequality for $\sigma$-sub-Gaussian random variables:

$$\mathbb{P}\big[|\hat{\mu}_i(t) - f(x_i)| > r\big] \leq 2\exp\Big(-\frac{n_i r^2}{2\sigma^2}\Big). \quad (5)$$

*Step 2: Calibration of confidence radius.* Substituting $r =$

$r_i(t) = \sigma\sqrt{2\log(2N_tT/\delta)/n_i}$:

$$\mathbb{P}\big[|\hat{\mu}_i(t) - f(x_i)| > r_i(t)\big] \leq 2\exp\left(-\frac{n_i \cdot 2\sigma^2 \log(2N_tT/\delta)}{2\sigma^2 \cdot n_i}\right)$$
$$= 2\exp\big(-\log(2N_tT/\delta)\big) = \frac{\delta}{N_tT}. \tag{6}$$

**Step 3: Union bound.** Applying the union bound over all $i \in \{1, \ldots, N_t\}$ and $t \in \{1, \ldots, T\}$:

$$\mathbb{P}[\mathcal{E}^c] \leq \sum_{t=1}^{T}\sum_{i=1}^{N_t} \frac{\delta}{N_tT} \leq \sum_{t=1}^{T} \frac{\delta}{T} = \delta. \tag{7}$$

Hence $\mathbb{P}[\mathcal{E}] \geq 1 - \delta$. □

### C.2. Proof of Lemma 4.2 (UCB envelope is valid)

*Proof.* Fix any $x \in \mathcal{X}$. For any sampled point $x_i$, on the good event $\mathcal{E}$:

$$f(x_i) \leq \hat{\mu}_i(t) + r_i(t) = \text{UCB}_i(t). \tag{8}$$

By Lipschitz continuity of $f$:

$$f(x) \leq f(x_i) + L \cdot d(x, x_i) \leq \text{UCB}_i(t) + L \cdot d(x, x_i). \tag{9}$$

Since this holds for all sampled $i$, taking the minimum over $i$:

$$f(x) \leq \min_{i \leq N_t}\{\text{UCB}_i(t) + L \cdot d(x, x_i)\} = U_t(x). \tag{10}$$

□

### C.3. Proof of Lemma 4.3 (Envelope slack bound)

*Proof.* Fix any $x \in \mathcal{X}$ and any sampled point $x_i$.

**Step 1: Upper confidence bound.** On the good event $\mathcal{E}$, we have $\hat{\mu}_i(t) \leq f(x_i) + r_i(t)$. Therefore:

$$\text{UCB}_i(t) = \hat{\mu}_i(t) + r_i(t) \leq f(x_i) + 2r_i(t). \tag{11}$$

**Step 2: Lipschitz propagation.** By Assumption 2.1 (Lipschitz continuity):

$$f(x_i) \leq f(x) + L \cdot d(x, x_i). \tag{12}$$

**Step 3: Combining bounds.** Substituting the Lipschitz bound into Step 1:

$$\text{UCB}_i(t) + L\,d(x, x_i) \leq f(x_i) + 2r_i(t) + L\,d(x, x_i)$$
$$\leq f(x) + L\,d(x, x_i) + 2r_i(t) + L\,d(x, x_i)$$
$$= f(x) + 2\big(r_i(t) + L\,d(x, x_i)\big). \tag{13}$$

**Step 4: Taking the minimum.** Since the above holds for all $i$, taking the minimum over $i$ on both sides:

$$U_t(x) = \min_{i \leq N_t}\{\text{UCB}_i(t) + L\,d(x, x_i)\}$$
$$\leq f(x) + 2\min_i\{r_i(t) + L\,d(x, x_i)\} \tag{14}$$
$$= f(x) + 2\rho_t(x).$$

□

### C.4. Proof of Theorem 4.5

*Proof.* We show that any point in the active set must have function value close to optimal.

**Step 1: Lower certificate validity.** On $\mathcal{E}$, for any sampled point $x_i$:

$$\text{LCB}_i(t) = \hat{\mu}_i(t) - r_i(t) \leq f(x_i) \leq f^*. \tag{15}$$

Taking the maximum over all $i$:

$$\ell_t = \max_{i \leq N_t} \text{LCB}_i(t) \leq f^*. \tag{16}$$

**Step 2: Active set membership implies high UCB.** Let $x \in A_t$. By definition of the active set (2):

$$U_t(x) \geq \ell_t. \tag{17}$$

**Step 3: Applying the envelope bound.** By Lemma 4.3:

$$f(x) + 2\rho_t(x) \geq U_t(x) \geq \ell_t. \tag{18}$$

**Step 4: Rearranging to obtain the containment.** Solving for $f(x)$:

$$f(x) \geq \ell_t - 2\rho_t(x)$$
$$= f^* - (f^* - \ell_t) - 2\rho_t(x)$$
$$\geq f^* - (f^* - \ell_t) - 2\sup_{x' \in A_t} \rho_t(x')$$
$$= f^* - 2\Delta_t. \tag{19}$$

Hence $A_t \subseteq \{x : f(x) \geq f^* - 2\Delta_t\}$. □

### C.5. Proof of Theorem 4.6

*Proof.* We connect the active set volume to the margin condition via the containment theorem.

**Step 1: Bounding $\Delta_t$.** From Theorem 4.5, $A_t \subseteq \{x : f(x) \geq f^* - 2\Delta_t\}$ where:

$$\Delta_t = \sup_{x \in A_t} \rho_t(x) + (f^* - \ell_t). \tag{20}$$

For any $x \in A_t$:

$$\rho_t(x) = \min_i \{r_i(t) + L \cdot d(x, x_i)\}$$
$$\leq \max_{i:x_i \text{ active}} r_i(t) + L \cdot \sup_{x \in A_t} \min_i d(x, x_i)$$
$$= \beta_t + L\eta_t. \tag{21}$$

Therefore:

$$\Delta_t \leq \beta_t + L\eta_t + \frac{\gamma_t}{2}. \tag{22}$$

*Step 2: Applying the margin condition.* By Assumption 2.4, for any $\varepsilon > 0$:

$$\text{Vol}(\{x : f(x) \geq f^* - \varepsilon\}) \leq C \cdot \varepsilon^{d-\alpha}. \tag{23}$$

*Step 3: Combining the bounds.* Setting $\varepsilon = 2\Delta_t \leq 2(\beta_t + L\eta_t) + \gamma_t$:

$$\text{Vol}(A_t) \leq \text{Vol}(\{x : f(x) \geq f^* - 2\Delta_t\})$$
$$\leq C \cdot (2\Delta_t)^{d-\alpha}$$
$$\leq C \cdot \left(2(\beta_t + L\eta_t) + \gamma_t\right)^{d-\alpha}. \tag{24}$$

$\square$

### C.6. Proof of Lemma 4.9: Algorithm 1 Drives $\beta_t$, $\eta_t$, $\gamma_t$

*Proof.* We prove each of the three claims.

*(i) Confidence radius $\beta_t$.* The replication rule (line 6 of Algorithm 1) allocates $\lceil (r_i(t)/\beta_{\text{sched}}(t))^2 \rceil$ samples to any active point $x_i$ with $r_i(t) > \beta_{\text{sched}}(t)$. Since $r_i(t) = \sigma\sqrt{2\log(2N_tT/\delta)/n_i}$ and the allocation drives $n_i$ to at least $\lceil 2\sigma^2\log(2N_tT/\delta)/\beta_{\text{sched}}^2(t)\rceil$, we obtain $r_i(t) \leq \beta_{\text{sched}}(t)$ after the replication step. Hence $\beta_t = \max_{i:x_i \text{ active}} r_i(t) \leq \beta_{\text{sched}}(t) = \sigma\sqrt{2\log(2T^2/\delta)/t}$, giving $\beta_t = O(\sigma\sqrt{\log(T/\delta)/t})$.

*(ii) Covering radius $\eta_t$.* Suppose $A_t$ contains an uncovered ball $B(x_0, \eta)$ with $\min_i d(x_0, x_i) = \eta$ and $x_0 \in A_t$. Then

$$\text{score}(x_0) = U_t(x_0) - L \cdot \min_i d(x_0, x_i) = U_t(x_0) - L\eta.$$

For any well-covered point $x'$ with $\min_i d(x', x_i) \leq \eta/4$, we have $\text{score}(x') = U_t(x') - L \cdot \min_i d(x', x_i) \leq U_t(x') - L\eta/4$. Since $U_t$ is $L$-Lipschitz (envelope of $L$-Lipschitz upper bounds), $|U_t(x_0) - U_t(x')| \leq L\,d(x_0, x')$, but for distant $x'$, this allows $U_t(x_0) \geq U_t(x') - L\,d(x_0, x')$, which combined with the score gap implies the oracle (returning within factor $C$ of the supremum) selects a point within $C\eta$ of the uncovered region. Repeating this argument, after $N_t = O((\text{Vol}(A_t)/\eta^d))$ active locations within $A_t$, we have $\eta_t \leq C\eta$. Substituting $\text{Vol}(A_t) \leq C\varepsilon_t^{d-\alpha}$ from Theorem 4.6 yields the claim. Approximation enters only through the constant $C$, not the exponent.

*(iii) Gap proxy $\gamma_t$.* Since $\ell_t = \max_i \text{LCB}_i(t)$ and $\text{LCB}_i(t) = \hat{\mu}_i(t) - r_i(t)$ is non-decreasing as $n_i$ increases (with high probability), $\ell_t$ is non-decreasing in $t$. The replication rule concentrates samples on the best-found active point $x^{\text{best}}$, driving $r_{\text{best}}(t) \to 0$. Once $\eta_t$ has reached the support of the global maximizer (item (ii)), $\hat{\mu}_{\text{best}}(t) \to f(x^{\text{best}}) \to f^*$, so $\gamma_t = f^* - \ell_t \to 0$ at rate $O(\beta_{\text{sched}}(t))$. $\square$

### C.7. Proof of Theorem 4.10

*Proof.* We derive the sample complexity by analyzing the requirements for $\varepsilon$-optimality, drawing on Lemma 4.9.

*Step 1: Optimality condition.* To achieve simple regret $r_T \leq \varepsilon$, it suffices to ensure $\varepsilon_t = 2(\beta_t + L\eta_t) + \gamma_t \leq \varepsilon$. This requires $\beta_t \leq \varepsilon/6$, $\eta_t \leq \varepsilon/(6L)$, and $\gamma_t \leq \varepsilon/3$.

*Step 2: Covering the active set.* By the shrinkage theorem (Theorem 4.6), once $2(\beta_t + L\eta_t) + \gamma_t \leq \varepsilon$ we have $\text{Vol}(A_t) \leq C \cdot \varepsilon^{d-\alpha}$. To achieve covering radius $\eta = \varepsilon/(6L)$ over a region of volume $C\varepsilon^{d-\alpha}$, Lemma 4.9(ii) requires:

$$N_{\text{cover}} = O\left(\frac{\text{Vol}(A_t)}{\eta^d}\right) = O\left(\frac{C\varepsilon^{d-\alpha}}{(\varepsilon/(6L))^d}\right) = O(C^d L^d \varepsilon^{-\alpha}) \tag{25}$$

distinct sample locations, where the $C^d$ factor reflects the approximation factor of the oracle.

*Step 3: Samples per location.* By Lemma 4.9(i), to achieve $\beta_t \leq \varepsilon/6$ we need

$$n_i = O\left(\frac{\sigma^2 \log(T/\delta)}{\varepsilon^2}\right). \tag{26}$$

*Step 4: Total sample complexity.* Combining Steps 2 and 3:

$$T = N_{\text{cover}} \cdot n_i$$
$$= O(C^d L^d \varepsilon^{-\alpha}) \cdot O\left(\frac{\sigma^2 \log(T/\delta)}{\varepsilon^2}\right)$$
$$= \tilde{O}(C^d L^d \varepsilon^{-(2+\alpha)}). \tag{27}$$

The gap proxy $\gamma_t \leq \varepsilon/3$ follows from Lemma 4.9(iii) given the above coverage and confidence. $\square$

### C.8. Proof of Theorem 4.11

*Proof.* We construct a hard instance via a randomized reduction with explicit $\alpha$-dependence.

*Step 1: Hard instance construction.* Fix $\alpha \in (0, d]$, $L \geq 1$, and the target $\varepsilon$. Set $M = \lceil \varepsilon^{-\alpha} \rceil$ and choose $M$ centers $\{c_1, \ldots, c_M\}$ as a maximal $\varepsilon^{\alpha/d}$-packing of $[0,1]^d$ (which has $\Theta(\varepsilon^{-\alpha})$ points by standard volume arguments). For each $i^* \in [M]$, define:

$$f_{i^*}(x) = (1 - \varepsilon) + \varepsilon \cdot \max\left\{0, 1 - \frac{L\|x - c_{i^*}\|}{\varepsilon}\right\}. \tag{28}$$

This is a cone-shaped bump centered at $c_{i^*}$ peaking at $f_{i^*}(c_{i^*}) = 1$, decreasing linearly with slope $L$ until reaching the baseline value $1 - \varepsilon$ at radius $\varepsilon/L$, and constant outside.

*Step 2: Verifying near-optimality dimension equals exactly $\alpha$.* The set $\{x : f_{i^*}(x) \geq f^* - \varepsilon\}$ is exactly the ball of radius $\varepsilon/L$ around $c_{i^*}$. This ball has volume $\mathrm{Vol}(B_{\varepsilon/L}) = c_d(\varepsilon/L)^d$ where $c_d$ is the unit-ball constant in dimension $d$. With $M = \Theta(\varepsilon^{-\alpha})$ centers, the union of all candidate locations covers at most $M \cdot c_d(\varepsilon/L)^d = \Theta(\varepsilon^{d-\alpha})$ volume. Restricted to the actual instance (a single $i^*$), the near-optimal set has volume $\Theta((\varepsilon/L)^d) = \Theta(\varepsilon^d)$. The near-optimality dimension (smallest $\alpha'$ such that volume scales as $\varepsilon^{d-\alpha'}$) is exactly $\alpha' = \alpha$ when the prior over $i^*$ is uniform, since the typical instance has $\Theta(\varepsilon^d)$ near-optimal volume but the family contains $M = \Theta(\varepsilon^{-\alpha})$ such pockets, making the structure equivalent to a problem with $\alpha$-dimensional difficulty.

*Step 3: Lipschitz validity.* Within the bump, the gradient has magnitude $L$; outside, $f \equiv 1 - \varepsilon$ is constant; the boundary matches continuously. So each $f_{i^*}$ is globally $L$-Lipschitz.

*Step 4: Information-theoretic lower bound.* Let $P_{i^*}$ denote the observation distribution under $f_{i^*}$ with $T$ adaptive queries. To identify $i^*$ with probability $\geq 2/3$ (sufficient for $\varepsilon$-optimality, since the only $\varepsilon$-optimal points lie within radius $\varepsilon/L$ of $c_{i^*}$), Fano's inequality requires:

$$\sum_{i^*=1}^{M} \mathbb{E}[N_{i^*}] \cdot \mathrm{KL}(P_{i^*} \| P_0) \geq \log(M/3),$$

where $N_{i^*}$ is the number of queries inside the support of bump $i^*$ and $P_0$ is the null (constant $1 - \varepsilon$) distribution.

*Step 5: Per-query KL.* Inside the bump support (volume $\sim (\varepsilon/L)^d$), the mean shifts by at most $\varepsilon$. For $\sigma$-sub-Gaussian noise, $\mathrm{KL}(P_{i^*} \| P_0) \leq \varepsilon^2/(2\sigma^2)$ per query inside the support, and zero outside.

*Step 6: Combining via change-of-measure.* By symmetry, the algorithm cannot distinguish among the $M$ hypotheses without samples in the appropriate cell. Combining Steps 4-5:

$$T \cdot \frac{\varepsilon^2}{2\sigma^2} \geq \log(M/3) = \Theta(\log \varepsilon^{-\alpha}) = \Theta(\alpha \log(1/\varepsilon)).$$

Solving for $T$ and noting that the per-cell sample requirement is also $\Omega(\sigma^2/\varepsilon^2)$ summed over $M$ cells:

$$T = \Omega\big(M \cdot \sigma^2/\varepsilon^2\big) = \Omega\big(\varepsilon^{-\alpha} \cdot \varepsilon^{-2}\big) = \Omega\big(\varepsilon^{-(2+\alpha)}\big).$$

This establishes the lower bound is tight for each specific $\alpha \in (0, d]$. $\square$

## C.9. Proof of Theorem 4.13 (Volume-Based Stopping)

*Proof.* By Theorem 4.5, on $\mathcal{E}$, $A_T \subseteq \{x : f(x) \geq f^* - 2\Delta_T\}$. By Assumption 4.12, for any $\varepsilon' > 0$ such that the set $\{f \geq f^* - \varepsilon'\}$ is contained in $A_T$:

$$\mathrm{Vol}(A_T) \geq c'(\varepsilon')^{d-\alpha}.$$

Hence the largest $\varepsilon'$ for which the inclusion is guaranteed satisfies:

$$\varepsilon' \leq (\mathrm{Vol}(A_T)/c')^{1/(d-\alpha)}.$$

Writing $r_T = f^* - f(\hat{x}_T) \leq 2\Delta_T = 2(\sup_{x \in A_T} \rho_T(x)) + \gamma_T$ and using that the algorithm's output $\hat{x}_T$ is the empirical maximizer (which lies in $A_T$ on $\mathcal{E}$):

$$r_T \leq (\mathrm{Vol}(A_T)/c')^{1/(d-\alpha)} + \gamma_T/2.$$

The stopping rule follows by setting each term $\leq \varepsilon/2$. $\square$

## C.10. Proof of Theorem 5.1

*Proof.* We prove each claim separately.

*Proof of (1): Bounded doubling events.* Each doubling event multiplies $\hat{L}$ by 2. Starting from $\hat{L}_0 \leq L^*$:

$$\hat{L}_k = 2^k \hat{L}_0 \quad \text{after } k \text{ doublings.} \tag{29}$$

The algorithm stops doubling when $\hat{L} \geq L^*$, which requires:

$$2^K \hat{L}_0 \geq L^* \implies K \geq \log_2(L^*/\hat{L}_0). \tag{30}$$

Hence $K \leq \lceil \log_2(L^*/\hat{L}_0) \rceil$.

*Proof of (2): Final estimate accuracy.* A violation is detected when:

$$|\hat{\mu}_i - \hat{\mu}_j| - 2(r_i + r_j) > \hat{L} \cdot d(x_i, x_j). \tag{31}$$

On the good event $\mathcal{E}$:

$$|f(x_i) - f(x_j)| \leq |\hat{\mu}_i - \hat{\mu}_j| + 2(r_i + r_j). \tag{32}$$

If $\hat{L} \geq L^*$, then by Lipschitz continuity:

$$|f(x_i) - f(x_j)| \leq L^* \cdot d(x_i, x_j) \leq \hat{L} \cdot d(x_i, x_j), \tag{33}$$

so no violation can occur. Thus violations only occur when $\hat{L} < L^*$, and after all doublings complete, $\hat{L} \geq L^*$. Since we double (rather than increase by smaller factors), $\hat{L} \leq 2L^*$.

*Proof of (3): Sample complexity overhead.* Between doublings, CGP runs with either:

- Invalid $\hat{L} < L^*$ (before sufficient doublings): certificates may be incorrect, but each such phase has at most $O(T/K)$ samples before a violation triggers doubling.

- Valid $\hat{L} \geq L^*$ (after final doubling): CGP achieves $\tilde{O}(\varepsilon^{-(2+\alpha)})$ complexity by Theorem 4.10.

There are at most $K$ invalid phases, each contributing $O(T/K)$ samples. The final valid phase dominates, giving total complexity $T = \tilde{O}(\varepsilon^{-(2+\alpha)} \cdot K) = \tilde{O}(\varepsilon^{-(2+\alpha)} \cdot \log(L^*/\hat{L}_0))$. $\qquad\square$

## C.11. Certificate Validity Under Adaptive Lipschitz Estimation

**Lemma C.1** (Certificate validity once $\hat{L} \geq L^*$). *Assume the good event $\mathcal{E}$ holds. Fix any time $t$ at which the current estimate satisfies $\hat{L} \geq L^*$. Then the envelope constructed with $\hat{L}$ satisfies $f(x) \leq U_t(x)$ for all $x \in \mathcal{X}$, and consequently the active set*

$$A_t = \{x \in \mathcal{X} : U_t(x) \geq \ell_t\}$$

*contains $x^*$ and certifies that any $x \notin A_t$ is suboptimal (in the sense $U_t(x) < \ell_t \leq f^*$).*

*Proof.* If $\hat{L} \geq L^*$, then for all $x, y \in \mathcal{X}$ we have $|f(x) - f(y)| \leq L^* d(x,y) \leq \hat{L} d(x,y)$, i.e., $f$ is $\hat{L}$-Lipschitz. On $\mathcal{E}$, for each sampled point $x_i$, $f(x_i) \leq \mathrm{UCB}_i(t)$. Therefore for all $x$,

$$f(x) \leq f(x_i) + \hat{L} d(x, x_i) \leq \mathrm{UCB}_i(t) + \hat{L} d(x, x_i).$$

Taking the minimum over $i$ yields $f(x) \leq U_t(x)$ for all $x$. In particular, $U_t(x^*) \geq f^*$. Also $\ell_t = \max_i \mathrm{LCB}_i(t) \leq f^*$ on $\mathcal{E}$, hence $U_t(x^*) \geq \ell_t$ and so $x^* \in A_t$. Finally, if $x \notin A_t$, then $U_t(x) < \ell_t \leq f^*$, certifying $x$ cannot be optimal under $\mathcal{E}$. $\qquad\square$

*Remark* C.2 (Why pre-final certificates need not be valid). If $\hat{L} < L^*$, then the envelope may fail to upper-bound $f$ globally, and the rule $U_t(x) \geq \ell_t$ can (in principle) exclude near-optimal points. CGP-Adaptive therefore guarantees certificate validity only after the final doubling event ensures $\hat{L} \geq L^*$.

## C.12. Global Safety of Certified Restarts (No False Elimination)

We restate and prove the key safety property underlying certified restarts in CGP-TR.

**Lemma C.3** (No false certified restart for the region containing $x^*$). *Fix any trust region $\mathcal{T}^*$ such that $x^* \in \mathcal{T}^*$. On the good event $\mathcal{E}$, the certified restart condition*

$$u_t^{(\mathcal{T}^*)} := \max_{x \in \mathcal{T}^*} U_t(x) < \ell_t$$

*never holds. Hence, $\mathcal{T}^*$ is never restarted by the certified rule.*

*Proof.* On the good event $\mathcal{E}$, Lemma 4.2 implies $f(x) \leq U_t(x)$ for all $x \in \mathcal{X}$. Since $x^* \in \mathcal{T}^*$,

$$u_t^{(\mathcal{T}^*)} = \max_{x \in \mathcal{T}^*} U_t(x) \geq U_t(x^*) \geq f(x^*) = f^*.$$

Also, by definition $\ell_t = \max_i \mathrm{LCB}_i(t)$. On $\mathcal{E}$, $\mathrm{LCB}_i(t) \leq f(x_i) \leq f^*$ for every sampled point $x_i$, hence $\ell_t \leq f^*$. Therefore,

$$u_t^{(\mathcal{T}^*)} \geq f^* \geq \ell_t,$$

so the strict inequality $u_t^{(\mathcal{T}^*)} < \ell_t$ cannot occur on $\mathcal{E}$. $\qquad\square$

## C.13. Proof of Theorem 6.1

We first establish a key lemma showing that certified elimination is safe.

**Lemma C.4** (Certified elimination). *On the good event $\mathcal{E}$, for any trust region $\mathcal{T}_j$,*

$$u_t^{(j)} := \max_{x \in \mathcal{T}_j} U_t(x) < \ell_t \quad\Longrightarrow\quad \sup_{x \in \mathcal{T}_j} f(x) < f(x^*).$$

*Proof.* On $\mathcal{E}$, by Lemma 4.2, $f(x) \leq U_t(x)$ for all $x \in \mathcal{X}$. Also $\ell_t = \max_i \mathrm{LCB}_i(t) \leq f(x^*)$ on $\mathcal{E}$ since each $\mathrm{LCB}_i(t) \leq f(x_i) \leq f(x^*)$. Therefore

$$\sup_{x \in \mathcal{T}_j} f(x) \leq \sup_{x \in \mathcal{T}_j} U_t(x) = u_t^{(j)} < \ell_t \leq f(x^*),$$

which proves the claim. $\qquad\square$

**Proof of Theorem 6.1(1): No false restarts.** If $x^* \in \mathcal{T}^*$, then $u_t^{(\mathcal{T}^*)} = \max_{x \in \mathcal{T}^*} U_t(x) \geq U_t(x^*) \geq f(x^*)$ on $\mathcal{E}$. Also $\ell_t \leq f(x^*)$ on $\mathcal{E}$. Hence $u_t^{(\mathcal{T}^*)} \geq \ell_t$ and the restart condition $u_t^{(\mathcal{T}^*)} < \ell_t$ never triggers.

**Proof of Theorem 6.1(2): Local certificate.** Conditioned on $T^*$ evaluations within $\mathcal{T}^*$, the CGP analysis applies verbatim on the restricted domain $\mathcal{T}^*$: the good event $\mathcal{E}$ implies all within-region confidence bounds hold; Lipschitz continuity holds on $\mathcal{T}^*$; and Assumption 2.4 holds restricted to $\mathcal{T}^*$. Therefore the containment and shrinkage results follow with $\mathcal{X}$ replaced by $\mathcal{T}^*$, yielding $\mathrm{Vol}(A_T^{(\mathcal{T}^*)}) \leq C\varepsilon^{d-\alpha}$ after $T^* = \tilde{O}(\varepsilon^{-(2+\alpha)})$ within-region samples.

**Proof of Theorem 6.1(3): Allocation bound.** Fix a suboptimal region $j$ with gap $\Delta_j > 0$. On $\mathcal{E}$, using the envelope bound from Lemma 4.3, we have for all $x$:

$$U_t(x) \leq f(x) + 2\rho_t(x),$$

where $\rho_t(x) = \min_i\{r_i(t) + Ld(x, x_i)\}$. Restricting to points sampled in $\mathcal{T}_j$ and using the within-region replication schedule, we bound $\rho_t(x) \leq \beta_j(n) + L \cdot \mathrm{diam}(\mathcal{T}_j)$, yielding

$$u_t^{(j)} = \max_{x \in \mathcal{T}_j} U_t(x) \leq \sup_{x \in \mathcal{T}_j} f(x) + 2\beta_j(n) + 2L \cdot \mathrm{diam}(\mathcal{T}_j),$$

after $n$ within-region samples. By the diameter condition $L \operatorname{diam}(\mathcal{T}_j) \leq \Delta_j/8$ and by requiring $\beta_j(n) \leq \Delta_j/8$, we obtain

$$u_t^{(j)} \leq \sup_{\mathcal{T}_j} f + \Delta_j/4 + \Delta_j/4 = f^* - \Delta_j/2.$$

Meanwhile, on $\mathcal{E}$ we have $\ell_t \leq f^*$ always, and once the region containing $x^*$ has been sampled sufficiently (which occurs because it is never restarted and is favored by UCB selection), $\ell_t$ becomes at least $f^* - \Delta_j/4$. Hence eventually $u_t^{(j)} < \ell_t$, triggering certified restart/elimination.

Solving $\beta_j(n) \leq \Delta_j/8$ under $\beta_j(n) \leq c_\sigma \sqrt{\log(c_T/\delta)/n}$ gives

$$n \geq \frac{64 c_\sigma^2}{\Delta_j^2} \log\left(\frac{c_T}{\delta}\right),$$

yielding the stated bound $N_j \leq \frac{64 c_\sigma^2}{\Delta_j^2} \log(c_T/\delta) + 1$.

### C.14. Proof of Proposition 7.1

*Proof.* We analyze each case and the certificate validity separately.

*Proof of (1): High smoothness ratio case.* When $\rho \geq 0.5$, CGP-Hybrid continues with CGP in Phase 2. The algorithm is identical to vanilla CGP, so by Theorem 4.10:

$$T = \tilde{O}\left(\varepsilon^{-(2+\alpha)}\right). \tag{34}$$

*Proof of (2): Low smoothness ratio case.* When $\rho < 0.5$, the function is significantly smoother near the optimum. After Phase 1, CGP has reduced the active set to volume $V < 0.1 \cdot \operatorname{Vol}(\mathcal{X})$. In Phase 2, GP-UCB operates within $A_t$, which has:

- Effective diameter $\operatorname{diam}(A_t) = O(V^{1/d})$.

- Local Lipschitz constant $L_{\text{local}} = \rho \cdot L < 0.5 L$.

The sample complexity of GP-UCB on this restricted domain depends on the kernel's maximum information gain $\gamma_T$ over $A_t$ (Srinivas et al., 2010). For commonly used kernels (Matérn, SE), $\gamma_T$ scales polylogarithmically with $T$ when the domain is bounded. The reduced diameter $O(V^{1/d})$ and local smoothness $\rho < 0.5$ empirically yield faster convergence than continuing CGP; we validate this empirically in Section 9 rather than claiming a specific rate.

*Proof of (3): Certificate validity.* The certificate $A_T$ is computed in Phase 1 using CGP's Lipschitz envelope construction. By Theorem 4.5, on the good event $\mathcal{E}$:

$$x^* \in A_T \quad \text{with probability} \geq 1 - \delta. \tag{35}$$

This guarantee depends only on the Lipschitz assumption and confidence bounds, not on the Phase 2 optimization

method. Therefore, switching to GP-UCB in Phase 2 does not invalidate the certificate: $A_T$ still contains $x^*$ with high probability, and any point outside $A_T$ remains certifiably suboptimal. $\qquad \square$

## D. Adaptive Noise Variance Estimation

When $\sigma$ is unknown, we extend the doubling principle from $L$ to $\sigma$. At each active point $x_i$ with $n_i \geq 5$ replications, compute the sample variance $\hat{\sigma}_i^2 = \frac{1}{n_i-1} \sum_{j=1}^{n_i} (y_{i,j} - \hat{\mu}_i)^2$. Aggregate: $\hat{\sigma}_t^2 = \max_i \hat{\sigma}_i^2$. By standard concentration, $\hat{\sigma}_t \leq c_1 \sigma \sqrt{\log(t/\delta)/n_i}$ with high probability for some constant $c_1$. Use $\hat{\sigma}_t \cdot (1 + c_2 \sqrt{\log(t/\delta)/n_i})$ as the calibrated estimate replacing $\sigma$ in the confidence radius formula. If a doubling event is detected on $\sigma$ (analogous to the $L$-doubling rule), increase $\hat{\sigma}$ by factor 2. The same $O(\log T)$ multiplicative overhead applies.

**Empirical validation.** On Hartmann-6 with $\sigma = 0.1$ but treated as unknown:

| Configuration | Regret ($\times 10^{-2}$) | Overhead |
|---|---|---|
| Known $\sigma$ (oracle) | $2.9 \pm 0.1$ | $1.0\times$ |
| Adaptive $\sigma$ (this method) | $3.1 \pm 0.1$ | $1.07\times$ |

Only $\sim 7\%$ regret degradation. The replication mechanism naturally provides repeated observations needed for variance estimation, so no additional queries are required.

## E. Smoothness Detection Stress Tests

We stress-test CGP-Hybrid's smoothness-detection heuristic on the adversarial example

$$f(x) = -\|x\|_2 + 0.5 \cdot \mathbf{1}[\|x\|_2 < 0.05],$$

which is globally smooth except for a sharp narrow peak at $x^* = 0$. The peak width (0.05) is below typical covering radii $\eta_t$ during Phase 1, so naive smoothness detection might incorrectly trigger GP refinement.

**Results.** Phase 1 of CGP-Hybrid concentrates $A_t$ near $x^*$ before estimating $\rho_t$. Once $A_t$ has shrunk sufficiently, the local Lipschitz estimate captures the spike:

| Method | Regret ($\times 10^{-2}$) | Stayed in CGP? |
|---|---|---|
| CGP-Hybrid ($\rho_{\text{thresh}} = 0.5$) | $1.4 \pm 0.1$ | Yes ($\hat{\rho} = 0.91$) |
| CGP-Hybrid ($\rho_{\text{thresh}} = 0.3$) | $1.5 \pm 0.1$ | Yes ($\hat{\rho} = 0.91$) |
| GP-UCB (kernel mismatch) | $4.2 \pm 0.3$ | N/A |
| Vanilla CGP | $1.4 \pm 0.1$ | N/A |

**Threshold sensitivity.** We vary $\rho_{\text{thresh}}$ on Hartmann-6 (where $\hat{\rho} = 0.72$ is borderline):

| $\rho_{\text{thresh}}$ | Regret ($\times 10^{-2}$) |
|---|---|
| 0.3 | $2.7 \pm 0.1$ |
| 0.5 (default) | $2.7 \pm 0.1$ |
| 0.7 | $2.8 \pm 0.1$ |

Variation under $5\%$ across this range, confirming the heuristic is robust.

**Safety fallback.**   Even if the heuristic misfires, Proposition 7.2 guarantees the certificate $A_T$ remains valid. Only convergence speed is affected, not correctness. Users in safety-critical applications can disable switching ($\rho_{\text{thresh}} = 0$) while preserving all certificate guarantees.

## F. Full Baseline Comparison

Table 7 presents the full $9 \times 12$ benchmark comparison referenced in Section 9. StoSOO and SAASBO did not converge on certain low-dimensional or high-dimensional problems and are marked accordingly.

**Discussion.**   StoSOO underperforms HOO by 10–20% on low-dimensional tasks at $T = 200$, consistent with its more conservative tree expansion. SAASBO is designed for high-dimensional smooth problems and does not converge on Needle-2D ($d = 2$ is below its sparsity regime). LIPO's online $L$ estimation degrades when underestimated; CGP-Adaptive's doubling scheme is provably better, as shown in Table 5.

## G. Practical Guidance

### G.1. Adaptive Lipschitz Estimation

CGP-Adaptive removes the requirement for known $L$ with $O(\log T)$ overhead. The doubling scheme is conservative but provably correct; more aggressive schemes (e.g., multiplicative updates with factor 1.5) may reduce overhead but risk certificate invalidation.

We recommend initializing $\hat{L}_0$ from finite differences on initial Sobol samples:

$$\hat{L}_0 = \max_{i \neq j} \frac{|y_i - y_j|}{d(x_i, x_j)}$$

over the first 10 samples. This typically underestimates $L$ by a factor of 2 to 10, requiring 1 to 4 doublings to reach $\hat{L} \geq L^*$. For users requiring anytime certificate validity, scale this initial estimate by $\sqrt{2}$ or larger; in 94% of our benchmarks this yields $\hat{L}_0 \geq L^*$ from iteration 1.

### G.2. Trust Region Configuration

CGP-TR trades global certificates for scalability. The local certificates within trust regions still enable principled stopping and progress assessment, but do not guarantee global optimality.

Recommended settings:

- Number of trust regions: $n_{\text{trust}} = 5$ (balances exploration vs. overhead)

- Initial radius: $r_0 = 0.2$ (covers 20% of domain diameter per region)

- Minimum radius: $r_{\min} = 0.01$ (prevents over-contraction)

- Failure threshold: $\tau_{\text{fail}} = 10$ (triggers contraction after 10 non-improving samples)

For applications requiring global certificates in high dimensions, combining CGP-TR with random embeddings (Wang et al., 2016) is promising: project to a low-dimensional subspace, run CGP with global certificates, then lift back.

### G.3. Smoothness Detection

CGP-Hybrid's smoothness detection via $\rho = L_{\text{local}}/L_{\text{global}}$ is heuristic but effective. We estimate $L_{\text{local}}$ from points within $A_t$ using the same finite difference approach as $L_{\text{global}}$.

The threshold $\rho_{\text{thresh}} = 0.5$ was selected via cross-validation on held-out benchmarks. More sophisticated detection could use local GP posterior variance or curvature estimates. The key insight is that CGP's certificate remains valid regardless of Phase 2 method, so switching is always safe. Appendix E validates robustness on adversarial inputs.

### G.4. When to Use CGP

CGP is well suited when:

1. Evaluations are expensive and interpretable progress is valued

2. The objective has margin structure (sharp peak rather than wide plateau)

3. Lipschitz continuity is a reasonable assumption

4. Dimension is moderate ($d \leq 15$ for vanilla CGP, $d \leq 100$ for CGP-TR)

5. Anytime stopping decisions are needed

For very high-dimensional problems ($d > 100$), trust region methods like TuRBO may scale better. For smooth problems with cheap evaluations, GP-based methods may be more sample efficient due to their ability to exploit higher-order smoothness.

*Table 7.* Full baseline comparison across all 12 benchmarks. Simple regret ($\times 10^{-2}$) at $T = 200$ (low/mid-$d$) or $T = 500$ (high-$d$); — indicates non-convergence or intractability. Bold: best per benchmark.

| Method | Needle | Branin | Hartmann | Levy | Rosen. | Ackley | SVM | LLand. | Rover-60 | Ant-100 |
|---|---|---|---|---|---|---|---|---|---|---|
| Random | 8.2 | 12.1 | 15.3 | 18.7 | 14.2 | 22.4 | 11.2 | 18.4 | 42.1 | 51.2 |
| GP-UCB | 2.1 | 1.8 | 4.2 | 5.1 | 4.8 | 12.3 | 3.9 | 8.2 | — | — |
| TuRBO | 1.8 | 2.1 | 3.1 | 4.3 | 3.9 | 9.8 | 3.2 | 7.4 | 12.4 | 18.7 |
| HEBO | 1.9 | 1.6 | 3.3 | 4.1 | 3.7 | 9.4 | 3.1 | 7.0 | 14.1 | 21.3 |
| BORE | 2.0 | 1.9 | 3.5 | 4.5 | 4.1 | 10.1 | 3.4 | 7.6 | 16.3 | 22.8 |
| HOO | 3.4 | 5.2 | 8.7 | 9.8 | 8.1 | 14.2 | 7.1 | 11.3 | — | — |
| StoSOO | 4.1 | 6.0 | 9.4 | 11.2 | 9.4 | 15.9 | 8.0 | 13.0 | — | — |
| LIPO | 2.8 | 4.3 | 6.2 | 7.1 | 5.9 | 13.4 | 5.2 | 9.8 | — | — |
| SAASBO | — | 2.0 | 3.4 | 4.4 | 3.8 | 9.6 | 3.3 | 7.3 | 13.2 | 19.8 |
| CGP | 1.2 | 2.0 | 2.9 | 3.8 | 3.8 | 8.1 | 2.8 | 6.4 | — | — |
| CGP-Adaptive | 1.3 | 2.1 | 3.0 | 3.9 | 3.9 | 8.3 | 2.9 | 6.5 | — | — |
| CGP-TR | 1.2 | 2.0 | 2.9 | 3.9 | 3.8 | 8.2 | 2.8 | 6.5 | **11.2** | **17.1** |
| **CGP-Hybrid** | **1.1** | **1.4** | **2.7** | **3.5** | **3.4** | **7.8** | **2.6** | **6.1** | **11.2** | **17.1** |

*Table 8.* Ablation study on Hartmann-6. Note: on Hartmann-6, $\hat{\rho} = 0.72 > 0.5$, so CGP-Hybrid stays in CGP; the small "$-$ GP refinement" contribution here reflects aggregate impact across all 12 benchmarks where GP is triggered.

| Variant | Regret ($\times 10^{-2}$) | Vol($A_{200}$) |
|---|---|---|
| CGP-Hybrid (full) | **2.7 ± 0.1** | 2.1% |
| − GP refinement | 2.9 ± 0.1 | 2.1% |
| − pruning certificate | 4.8 ± 0.2 | – |
| − coverage penalty | 3.9 ± 0.2 | 3.8% |
| − replication | 3.6 ± 0.2 | 2.9% |
| CGP-TR ($d = 6$) | 2.8 ± 0.1 | 2.4% (local) |
| CGP-Adaptive | 3.0 ± 0.1 | 2.4% |

*Table 9.* Certificate-enabled early stopping on Hartmann-6. The certified rule uses $\varepsilon_t$ from (4); Vol($A_t$)-based stopping is valid under Assumption 4.12.

| Stopping Rule | Samples | Regret ($\times 10^{-2}$) | Savings |
|---|---|---|---|
| Fixed $T = 200$ | 200 | 2.9 ± 0.1 | – |
| $\varepsilon_t \leq 0.05$ (certified) | 134 ± 12 | 3.0 ± 0.1 | 33% |
| Vol($A_t$) < 10% | 82 ± 9 | 3.8 ± 0.2 | 59% |
| Vol($A_t$) < 5% | 118 ± 14 | 3.2 ± 0.2 | 41% |

# H. Computational Performance and Empirical Validation of $\alpha$

This appendix reports two empirical validations referenced from the main paper.

**Wall-clock performance.** Table 10 compares per-run wall-clock time on Hartmann-6 with $T = 200$. CGP variants are 6–8× faster than GP-based methods, consistent with their $O(N_t)$ per-iteration envelope cost versus GP's $O(n^3)$ posterior update. The minor overhead of CGP-Adaptive and CGP-Hybrid relative to vanilla CGP comes from doubling-event checking and GP refinement, respectively.

*Table 10.* Wall-clock time (seconds) for $T = 200$ on Hartmann-6.

| Method | Time (s) | Regret ($\times 10^{-2}$) | Speedup |
|---|---|---|---|
| CGP | 58 | 2.9 | 8× |
| CGP-Adaptive | 64 | 3.0 | 7.5× |
| CGP-Hybrid | 72 | 2.7 | 6.7× |
| GP-UCB | 480 | 4.2 | 1× |
| TuRBO | 620 | 3.1 | 0.8× |
| HEBO | 890 | 3.3 | 0.5× |

**Empirical $\hat{\alpha}$ estimates.** Table 11 reports the near-optimality dimension estimated from observed shrinkage trajectories: we fit $\log \text{Vol}(A_t) = a + (d - \hat{\alpha}) \log \varepsilon_t$ via least squares using monitoring samples of $\varepsilon_t$ and Vol($A_t$). Across all benchmarks, $\hat{\alpha} < d$, confirming that the margin condition (Assumption 2.4) holds non-trivially and our $\tilde{O}(\varepsilon^{-(2+\alpha)})$ rate strictly improves over the worst case $\tilde{O}(\varepsilon^{-(2+d)})$.

*Table 11.* Empirical $\hat{\alpha}$ estimates from shrinkage trajectories. 95% CI from bootstrap over 30 runs.

| Benchmark | $d$ | $\hat{\alpha}$ (95% CI) | True $\alpha$ | $\hat{\alpha} < d$? |
|---|---|---|---|---|
| Needle-2D | 2 | 1.8 ± 0.2 | 2.0 | ✓ |
| Branin | 2 | 1.2 ± 0.1 | – | ✓ |
| Hartmann-6 | 6 | 2.4 ± 0.3 | – | ✓ |
| Ackley-10 | 10 | 3.2 ± 0.4 | – | ✓ |
| Rover-60 | 60 | 8.4 ± 1.2 | – | ✓ |

# I. Experimental Details

## I.1. Baseline Configurations

- **Random Search**: Sobol sequences for quasi-random sampling

- **GP-UCB**: Matérn-5/2 kernel via BoTorch, $\beta_t = 2 \log(t^2 \pi^2 / 6\delta)$

- **TuRBO**: Default settings from Eriksson et al. (2019), 1 trust region

- **HEBO**: Heteroscedastic GP with input warping, default settings

- **BORE**: Tree-Parzen estimator with density ratio, default settings

- **HOO**: Binary tree with $\nu_1 = 1$, $\rho = 0.5$

- **StoSOO**: $k = 3$ children per node, $h_{\max} = 20$

- **LIPO**: Pure Lipschitz optimization, $L$ estimated online

- **SAASBO**: Sparse axis-aligned GP, 10 active dimensions

### I.2. Benchmark Details

**Low-dimensional.**

- **Needle-2D**: $f(x) = 1 - \|x - x^*\|^{1/\alpha}$ with $\alpha = 2$, sharp peak

- **Branin**: Standard 2D benchmark with 3 global optima

- **Hartmann-6**: 6D benchmark with narrow global basin

- **Levy-5**: 5D benchmark with global structure

- **Rosenbrock-4**: 4D benchmark with curved valley

**Medium-dimensional.**

- **Ackley-10**: 10D benchmark with many local optima

- **SVM-RBF-6**: Real hyperparameter tuning ($C$, $\gamma$, 4 preprocessing) on MNIST

- **LunarLander-12**: RL reward optimization with 12 policy parameters

**High-dimensional.**

- **Rover-60**: Mars rover trajectory with 60 waypoint parameters (Wang et al., 2016)

- **NAS-36**: Neural architecture search on CIFAR-10, 36 continuous encodings

- **Ant-100**: MuJoCo Ant locomotion, 100 morphology and control parameters

### I.3. Computational Resources

All experiments run on AMD EPYC 7763 with 256GB RAM. CGP variants use NumPy/SciPy; GP baselines use BoTorch/GPyTorch with GPU acceleration (NVIDIA A100) where available.

