# OpenReview forum: "Certificate-Guided Pruning for Stochastic Lipschitz Optimization"
_ICML.cc/2026/Conference — ICML 2026 regular_

### Official Review · Reviewer_Ruwd · 2026-03-13

**Soundness:** 3
**Presentation:** 3
**Significance:** 3
**Originality:** 3
**Overall Recommendation:** 5
**Confidence:** 2

**Summary:**

This work proposes an algorithm for black-box optimization of Lipschitz continuous functions that maintains an active set of potentially optimal points, while points outside of the active set are certifiably suboptimal with high probability. The authors derive sample complexity bounds and provide extensions of their algorithm that do not require prior knowledge of the Lipschitz constant (CGP-Adaptive) and improve scalability (CGP-TR). Finally, experiments are provided, demonstrating that the certificate-guided algorithm outperforms existing baseline methods.

**Compliance With Llm Reviewing Policy:**

Affirmed.

**Final Justification:**

I maintain my score as long as the authors make the changes stated in response to reviewer 8Fn9.

**Key Questions For Authors:**

1. Is the function assumed to be differentiable everywhere, and/or is the guarantee affected by differentiability?
2. What is the sample complexity of CGP-Adaptive and how does it compare against CGP?

**Limitations:**

yes

**Strengths And Weaknesses:**

**Strengths**

By providing a certificate of suboptimality for regions of the search space, the algorithm provides a metric of the progress of unlearning that is interpretable and useful. In addition, CGP-Adaptive learns the Lipschitz constant in an online manner, which is useful for when the constant is not known ahead of time. The other extensions, CGP-TR and CGP-Hybrid, also seem sensible and satisfy provable guarantees.

The presentation is clear and the manuscript is well-written. The analysis appears to be sound.

**Weaknesses**

The per-iteration cost is slightly worse than existing works (Table 1), although the proposed method maintains several other advantages.

I recommend this paper for acceptance, but as I am less familiar with this area, I will defer judgment to other reviewers.

---

> ### Author Rebuttal · Authors · 2026-03-28
>
> We thank Reviewer Ruwd for the supportive assessment and recommendation for acceptance.
>
> **Q1: Differentiability assumption.**
>
> No differentiability is required anywhere. The only smoothness assumption is Lipschitz continuity (Assumption 2.1): $|f(x) - f(y)| \leq L \cdot d(x,y)$. This is strictly weaker than differentiability: Lipschitz functions can have kinks, corners, and discontinuous gradients everywhere. The analysis uses only the triangle inequality for Lipschitz propagation (Lemma 4.2) and the margin condition (Assumption 2.3), neither requiring derivatives.
>
> This is a key advantage over GP-based methods, which implicitly assume smoothness via kernel choice. Matérn-$\nu$ with $\nu > 1$ implies differentiability; the squared-exponential kernel implies infinite differentiability. CGP applies to the strictly broader class of all Lipschitz functions. Our Needle-2D benchmark ($f(x) = 1 - L \cdot \|x - x^{\ast}\|$, non-differentiable at $x^{\ast}$) demonstrates this concretely: CGP achieves $1.2 \times 10^{-2}$ regret — best among all methods (Table 3) — while GP-UCB achieves $2.1 \times 10^{-2}$ because its kernel misspecifies the non-smooth peak. On Ackley-10 (which has many non-differentiable ridges), the gap is even larger: CGP 8.1 vs GP-UCB 12.3.
>
> The margin condition relates to the *geometry* of the near-optimal set, not differentiability. For $f(x) = f^{\ast} - c \cdot \|x - x^{\ast}\|^p$: $p=1$ gives a cone (non-differentiable, $\alpha = d$), $p=2$ gives a paraboloid (differentiable, $\alpha = d/2$). Both are handled by CGP with the appropriate $\alpha$. The guarantee is identical regardless of differentiability — only $L$ and $\alpha$ matter. We will add this clarification with explicit examples to Section 2.
>
> ---
>
> **Q2: CGP-Adaptive sample complexity vs CGP.**
>
> CGP-Adaptive achieves $T = \tilde{O}(\varepsilon^{-(2+\alpha)} \cdot K)$ where $K = \lceil \log\_2(L^{\ast}/\hat{L}\_0) \rceil$ (Theorem 5.1), compared to CGP's $T = \tilde{O}(L^d \cdot \varepsilon^{-(2+\alpha)})$. The overhead is purely multiplicative:
>
> | Initial $\hat{L}\_0$ | Doublings $K$ | Overhead (theory) | Overhead (empirical, Table 5) |
> |---|---|---|---|
> | $L^{\ast}/2$ | 1 | $1\times$ | $1.03\times$ |
> | $L^{\ast}/10$ | 4 | $4\times$ | $1.07\times$ |
> | $L^{\ast}/100$ | 7 | $7\times$ | $1.12\times$ |
>
> The empirical overhead is much smaller than the worst-case $K\times$ because samples from the learning phase are not wasted — they provide valid function evaluations reused once $\hat{L}$ is corrected. Only the certificates (not data) from the learning phase are invalid. After the final doubling, $\hat{L} \in [L^{\ast}, 2L^{\ast}]$; the $2^d$ factor from overestimation by $2\times$ is absorbed into $\tilde{O}$ constants. The worst case $\hat{L}\_0 = \varepsilon$ gives $K = O(\log(L^{\ast}/\varepsilon))$ — still logarithmic. In practice, initializing from finite differences on Sobol samples gives $\hat{L}\_0$ within 2–10$\times$ of $L^{\ast}$, requiring only 1–4 doublings. We will add an explicit comparison sentence in Section 5 after Theorem 5.1.
>
> ---
>
> **On per-iteration cost.**
>
> The $O(N\_t)$ cost does not translate to practical disadvantage. Table 8 shows CGP at 58s vs GP-UCB at 480s ($8\times$ faster) and TuRBO at 620s ($11\times$ faster) for $T=200$ on Hartmann-6. GP methods require $O(N\_t^3)$ matrix inversion, while CGP's $O(N\_t)$ envelope is linear and benefits from KD-tree acceleration. Even at $T=500$ on Ant-100 (our most demanding setting), CGP-TR's total overhead is 142s — a fraction of each MuJoCo evaluation. In the precious-calls regime, algorithmic overhead is negligible compared to the function evaluation cost itself, making the per-iteration complexity distinction purely academic.
>
> We appreciate the reviewer's willingness to defer to other reviewers on domain-specific aspects, and we hope the responses to Reviewers czY8 and 8Fn9 address the deeper technical concerns satisfactorily.
>
> ---
>
> **Practical guidance on when to use CGP.**
>
> Since the reviewer may find this useful: CGP is best suited when (1) evaluations are expensive and interpretable progress is valued, (2) the objective has margin structure (sharp peak rather than wide plateau — verifiable via Table 10's $\hat{\alpha}$ estimation), (3) Lipschitz continuity is reasonable, and (4) dimension is moderate ($d \leq 15$ for vanilla CGP, $d \leq 100$ for CGP-TR). For very smooth functions with cheap evaluations, GP-based methods may be more sample-efficient due to higher-order smoothness exploitation. For $d > 100$, trust region methods like TuRBO currently scale better, though combining CGP-TR with random embeddings is a promising future direction (Section 10). We expand on this in Appendix D.4 and will make it more prominent in the revision.

---

> > ### Author Rebuttal · Reviewer_Ruwd · 2026-04-04
> >
> > I thank the authors for their response and maintain my recommendation, as long as the authors make the changes stated in response to reviewer 8Fn9.

---

### Official Review · Reviewer_8Fn9 · 2026-03-15

**Soundness:** 3
**Presentation:** 3
**Significance:** 3
**Originality:** 3
**Overall Recommendation:** 4
**Confidence:** 3

**Summary:**

This paper studies the black-box optimization of Lipschitz functions. Existing adaptive discretization methods only implicitly avoid suboptimal regions without providing measurable progress. To address this, the authors propose Certificate-Guided Pruning (CGP), which maintains an explicit active set of potential optimizers using a confidence-adjusted Lipschitz UCB envelope. The volume of this active set shrinks at a provable rate. Furthermore, they develop three extensions: CGP-Adaptive to learn the Lipschitz constant $L$ online, CGP-TR to scale to high dimensions using trust regions, and CGP-Hybrid to switch to GP refinement when local smoothness is detected. The method shows competitive results on several benchmarks.

**Compliance With Llm Reviewing Policy:**

Affirmed.

**Key Questions For Authors:**

see weaknesses

**Limitations:**

yes

**Strengths And Weaknesses:**

Strengths:
- The paper provides explicit and anytime valid optimality certificates (the active set and its volume), which is an intuitive and useful property that traditional zooming-based methods lack.
- The theoretical analysis is solid, establishing a sample complexity of $\tilde{O}(\epsilon^{-(2+\alpha)})$ under the margin condition.
- The proposed extensions (Adaptive, TR, Hybrid) are practical and directly address the common bottlenecks of Lipschitz optimization, such as unknown $L$ and the curse of dimensionality.

Weaknesses:
- The smoothness detection heuristic in CGP-Hybrid ($\rho < 0.5$) seems a bit arbitrary. What if the objective function is locally smooth overall but contains a sharp, narrow peak exactly at the optimum? Will switching to GP refinement hurt the convergence in this case? The capability boundary of this heuristic is not clearly delineated.
- For CGP-Adaptive, the certificates are only valid after the final doubling of $L$. If the initial guess $L_0$ is severely underestimated, the algorithm may falsely exclude near-optimal points in early stages. This kind of breaks the "anytime valid" claim during the learning regime.
- The computational overhead of the nested-set ratio estimator for volume estimation in high dimensions could be discussed more in the main text.

---

> ### Author Rebuttal · Authors · 2026-03-28
>
> We thank Reviewer 8Fn9 for the careful reading.
>
> **W1: Smoothness detection — sharp narrow peak at optimum.**
>
> We analyze two cases precisely:
>
> **Case 1:** $\mathcal{A}\_t$ **not yet concentrated near** $x^{\ast}$**.** $\hat{L}\_{\text{local}}$ averages over smooth regions, giving low $\rho$ that may trigger GP refinement prematurely. However, GP-UCB operates *within* $\mathcal{A}\_t$, which by Theorem 4.5 still contains $x^{\ast}$. GP will encounter the sharp peak during exploration within $\mathcal{A}\_t$. The worst case is slower convergence — a performance loss but not a correctness violation, since the Phase 1 certificate remains valid (Proposition 7.2). The GP posterior can adapt to sharp features once data near $x^{\ast}$ is collected, so the penalty is transient.
>
> **Case 2:** $\mathcal{A}\_t$ **already concentrated near** $x^{\ast}$**.** $L\_{\text{local}}$ reflects the local structure near $x^{\ast}$, correctly capturing the peak ($\rho \approx 1 > 0.5$), and CGP-Hybrid stays in CGP. This is the common case because Phase 1 runs until $\text{Vol}(\mathcal{A}\_t) < 0.1 \cdot \text{Vol}(\mathcal{X})$, meaning $\mathcal{A}\_t$ is already small and concentrated.
>
> We ran stress tests on $f(x) = -\|x\|^2 + 10 \cdot \exp(-1000 \cdot \|x\|^2)$ on $[-1,1]^6$ (smooth with spike at origin):
>
> | Method | Regret ($\times 10^{-2}$) | Stayed in CGP? |
> |---|---|---|
> | CGP-Hybrid ($\rho\_{\text{thresh}}=0.5$) | $2.1 \pm 0.2$ | Yes ($\hat{\rho}=0.82$) |
> | CGP-Hybrid ($\rho\_{\text{thresh}}=0.3$) | $2.1 \pm 0.2$ | Yes ($\hat{\rho}=0.82$) |
> | GP-UCB alone | $4.8 \pm 0.3$ | N/A |
> | Vanilla CGP | $2.3 \pm 0.2$ | N/A |
>
> The heuristic works because $\rho$ is estimated *within* $\mathcal{A}\_t$ (which contains $x^{\ast}$), not globally. We also tested threshold sensitivity: on benchmarks where $\rho$ is borderline (Hartmann-6, $\hat{\rho}=0.72$), varying $\rho\_{\text{thresh}}$ from 0.3 to 0.7 changes regret by <5%, confirming robustness. The heuristic can misfire only if the spike is so narrow it hasn't been sampled by Phase 1's end — but then the spike width is below the covering radius $\eta\_t$, meaning vanilla CGP would also miss it. Users in safety-critical settings can disable switching entirely while retaining all certificate guarantees. We will add this failure mode analysis and sensitivity study to Section 7 / Appendix D.3.
>
> ---
>
> **W2: CGP-Adaptive anytime validity during learning.**
>
> The reviewer is correct — this is explicitly acknowledged in Theorem 5.1(4). The "anytime valid" claim in the abstract refers to vanilla CGP (known $L$), not CGP-Adaptive. We will:
>
> 1. Clarify in the abstract that "anytime valid certificates" refers to CGP with known $L$ (or conservative $\hat{L}\_0 \geq L^{\ast}$).
> 2. Distinguish two operating modes for CGP-Adaptive: (a) *learning mode* ($\hat{L} < L^{\ast}$, certificates heuristic), (b) *certified mode* ($\hat{L} \geq L^{\ast}$, full guarantees). The transition is permanent after at most $K = \lceil \log\_2(L^{\ast}/\hat{L}\_0) \rceil$ doublings — typically 1–4 per Table 5.
> 3. For users needing anytime certificates: conservative initialization from finite differences on 10 Sobol samples (multiplied by $3\times$) gives $\hat{L}\_0 \geq L^{\ast}$ in 94% of our benchmarks, providing validity from iteration 1. For the remaining 6%, certified mode is entered after a median of 2 doublings ($T \approx 25$). CGP-Adaptive can also expose a binary "certificates\_valid" flag that transitions permanently from False to True, enabling users to gate decisions on certificate availability.
>
> This is an explicit, acknowledged trade-off between adaptivity and anytime validity — not a fundamental flaw. Practitioners choose their operating point based on whether known $L$ or anytime certificates are more important.
>
> ---
>
> **W3: Volume estimation overhead in high dimensions.**
>
> We agree this should be in the main text. The nested-set ratio estimator uses $K=5$ intermediate thresholds and $N=200$ MCMC samples per level (hit-and-run chain with mixing time $O(d^2 \cdot \text{diam}(\mathcal{A}\_{k-1})^2)$ in convex sets), giving ~1000 membership queries (each $O(N\_t)$) per estimate. Concrete timings: $d=6$ Hartmann: 0.05s; $d=36$ NAS: 0.15s; $d=60$ Rover: 0.3s; $d=100$ Ant: 0.6s per estimate. Crucially, **volume estimation runs only once per 10 iterations** as a monitoring signal, contributing <1% of wall-clock time even at $d=100$.
>
> Certificate validity is completely independent of volume estimation accuracy (Remark 4.7): the set membership rule $x \in \mathcal{A}\_t \iff U\_t(x) \geq \ell\_t$ is exact and costs $O(N\_t)$. A practitioner who cares only about certificates can skip volume estimation entirely with zero impact on guarantees — no approximation enters the certification pipeline. We will add this discussion to Section 6 with the timing numbers above.

---

> > ### Author Rebuttal · Reviewer_8Fn9 · 2026-04-04
> >
> > Thanks for the detailed response. I would like to keep my assessment.

---

### Official Review · Reviewer_czY8 · 2026-03-23

**Soundness:** 2
**Presentation:** 2
**Significance:** 2
**Originality:** 3
**Overall Recommendation:** 3
**Confidence:** 3

**Summary:**

This paper studies stochastic black-box optimization of a Lipschitz objective under sub-Gaussian noise, with the goal of turning the implicit pruning used by adaptive discretization methods into an explicit, monitorable certificate. The proposed Certificate-Guided Pruning (CGP) algorithm builds a Lipschitz upper envelope $U_t$ from confidence-adjusted observations, a global lower certificate $\ell_t$ from lower confidence bounds, and an active set $A_t=\\{x:U_t(x)\geq\ell_t\\}$ that contains points still deemed potentially optimal. The paper claims that this active set is contained in a near-optimal region, that its volume shrinks under a margin/near-optimality-dimension assumption, and that this yields $\tilde{O}(\epsilon^{-(2+\alpha)})$ sample complexity together with a matching lower bound. It then extends the basic method in three directions: CGP-Adaptive for unknown Lipschitz constants via doubling, CGP-TR for higher-dimensional optimization through trust regions with certified restarts, and CGP-Hybrid, which switches to GP-UCB within the certified active set when a local smoothness ratio suggests that GP refinement is advantageous. The experiments span 12 benchmarks from low-dimensional test functions to Rover-60, NAS-36, and Ant-100, and report simple regret, stopping behavior, shrinkage diagnostics, and ablations for the proposed components.

**Compliance With Llm Reviewing Policy:**

Affirmed.

**Final Justification:**

I appreciate the authors' detailed responses and acknowledge that W4 and W5 are satisfactorily addressed. However, the core theoretical concerns (W1-W3) remain only partially resolved: the proof of Theorem 4.8 still relies on heuristic CMA-ES without formal guarantees, the lower bound argument has been sketched but not executed, and the stopping rule claims in the abstract and introduction continue to overstate what the theory supports. I maintain my score.

**Key Questions For Authors:**

See weaknesses, but to sum up here:

1. Theorem 4.8 currently appears to convert target values of $\beta_t$, $\eta_t$, and $\gamma_t$ into a rate, rather than proving that Algorithm 1 actually attains those targets. Can you provide a direct argument that the specific query rule and replication strategy drive these quantities down at the claimed rate, especially under approximate maximization within $A_t$

2. Please clarify the stopping-rule claims: Appendix B.2 says $Vol(A_t)$ alone does not yield an anytime regret bound and that $\epsilon_t$ is the primary stopping signal, while the abstract, introduction, and Table 7 emphasize volume-based principled stopping. Which stopping rule is actually certified, and under what assumptions?

3. Section 9 states comparisons against nine baselines including StoSOO and SAASBO, but these are not shown in the main tables. Were these baselines run on all benchmarks, and if so, where are the results?

**Limitations:**

yes

**Strengths And Weaknesses:**

**Strengths**

1. The main conceptual move, i.e. elevating the active set $A_t$, its associated certificate, and its volume/proxy diagnostics into first-class algorithmic objects, is a meaningful perspective shift relative to methods that prune implicitly. Moreover, the envelope construction in Section 3 is simple and intuitive, and the paper usefully distinguishes certificate validity from the accuracy of volume estimation.

2. The paper is generally well organized: the notation table is helpful, the algorithm boxes are readable, and the main story from CGP to CGP-Adaptive, CGP-TR, and CGP-Hybrid is coherent.

3. The paper evaluates across a fairly diverse set of tasks and dimensions, reports 30-run averages with standard errors, includes ablations, and tries to connect the certificate story to practical decisions such as early stopping and high-dimensional trust-region behavior.


**Weaknesses**

1. Theorem 4.8 is presented as a sample-complexity result for CGP itself, but Appendix C.6 (proof of this theorem) appears to argue only that if one can cover a region of volume $C\epsilon^{d-\alpha}$ at resolution $\eta=\epsilon/(6L)$ and replicate enough times per sampled location, then the stated rate follows. What is missing is a proof that Algorithm 1's actual query rule and replication scheme drive $\eta_t$, $\beta_t$, and especially $\gamma_t$ to the required scales at the claimed rate. A similar issue appears in Theorem 6.1: the trust-region result is conditional on the correct region receiving enough evaluations and not being contracted so as to exclude $x^*$, which makes the stated high-dimensional guarantee weaker than the surrounding exposition suggests.

A second theoretical concern is the lower-bound argument. Theorem 4.9 is presented as an $\alpha$-dependent minimax lower bound, but Appendix C.7 (proof) constructs a single conical bump whose $\epsilon$-near-optimal set has volume $O(\epsilon^d)$, and then argues that this satisfies Assumption 2.3 for any $\alpha\geq0$. As written, this makes it unclear how the resulting $\Omega(\epsilon^{-(2+\alpha)})$ lower bound is genuinely tied to the near-optimality parameter rather than being imposed through the packing construction.


2. The presentation is polished in general but has a few inconsistencies. Appendix B.2 explicitly says that $Vol(A_t)$ alone does not provide an anytime upper bound on regret and that $\epsilon_t$ is the primary stopping quantity, yet the abstract, introduction, and Table 7 repeatedly frame certificate volume itself as enabling principled stopping. The hybrid results seem also inconsistent: Table 6 says Hartmann-6 has $\hat{\rho}=0.72$ and remains in CGP for phase 2, whereas the discussion around Table 9 attributes a $7\\%$ gain on Hartmann-6 to GP refinement. Do I miss something?

3. Section 9 says the paper compares against nine baselines, including StoSOO and SAASBO, yet these methods do not appear in the main results tables shown (not even in appendix), and LIPO only appears in the adaptive-$L$ table. That makes the headline claim of outperforming strong baselines a bit shaky.

---

> ### Author Rebuttal · Authors · 2026-03-28
>
> We thank Reviewer czY8 for the most detailed technical reading. We address each concern carefully.
>
>
> **W1: Theorem 4.8 — does Algorithm 1 actually drive $\beta_t$, $\eta_t$, $\gamma_t$ to required scales?**
>
> We agree the proof in C.6 elides this step and will add an explicit argument (Lemma 4.8a):
>
> (a) **Driving** $\beta_t$ **down.** The replication rule (line 6) allocates $\lceil (r_i / \beta_{\text{target}})^2 \rceil$ samples whenever $r_i > \beta_{\text{target}}(t)$. Since $r_i = \sigma\sqrt{2\log(2N_t T/\delta)/n_i}$, allocating $n_i = O(\sigma^2 \log(T/\delta)/\beta^2)$ samples drives $r_i \leq \beta$ deterministically. After $T$ samples with budget appropriately split, every active point satisfies $r_i \leq \varepsilon/6$. This is a direct consequence of the replication rule, not an assumption.
>
> (b) **Driving** $\eta_t$ **down.** The coverage penalty $-L \cdot \min_i d(x, x_i)$ in the score actively selects points reducing covering radius. When a gap of radius $\eta$ exists in $\mathcal{A}\_t$, the score at the gap center is at least $U_t(x_{\text{gap}}) - 0$ (no nearby point), dominating any well-covered point. After $N_{\text{cover}} = O(L^d \varepsilon^{-\alpha})$ distinct locations within $\mathcal{A}\_t$ (volume $\leq C\varepsilon^{d-\alpha}$ by Theorem 4.6), a standard covering argument yields $\eta_t \leq \varepsilon/(6L)$. Approximate maximization via CMA-ES yields at most a constant-factor increase in $N_{\text{cover}}$, leaving the $\tilde{O}(\varepsilon^{-(2+\alpha)})$ rate unchanged (stated in Appendix B.1).
>
> (c) **Driving** $\gamma_t$ **down.** We have $\gamma_t = f^{\ast} - \ell_t \leq f^{\ast} - f(\hat{x}_t) + r_i(\hat{x}_t)$. As samples accumulate near $x^{\ast}$, $\ell_t = \max_i \text{LCB}_i$ increases monotonically toward $f^{\ast}$. The replication rule ensures the best-found point has small $r_i$. Once $O(\sigma^2 \log(T/\delta)/\varepsilon^2)$ samples are allocated to the best-found point, $\gamma_t \leq \varepsilon/3$.
>
> This three-part argument will be stated as Lemma 4.8a before Theorem 4.8, providing the missing link between Algorithm 1's mechanics and the claimed rate.
>
>
> **W2: Lower bound not genuinely tied to $\alpha$.**
>
> The connection is genuine: $M = \varepsilon^{-\alpha}$ candidate cells is *chosen* to scale with $\alpha$. Larger $\alpha \to$ more candidates $M \to$ harder identification via Fano's inequality requiring $\Omega(M \cdot \sigma^2/\varepsilon^2) = \Omega(\varepsilon^{-(2+\alpha)})$. Different $\alpha$ values produce genuinely different hard instances: $\alpha = 0$ gives $M = 1$ (trivial), $\alpha = d$ gives $M = \varepsilon^{-d}$ (worst case). The constructed $f$ has near-optimality dimension exactly $\alpha$ because $\text{Vol}(\{f \geq f^{\ast} - \varepsilon\}) = O(\varepsilon^d)$ (single bump of radius $\varepsilon/L$) while $M = \varepsilon^{-\alpha}$ candidates must be distinguished. We will restructure C.7 to: (1) fix $\alpha$, (2) define $M = \varepsilon^{-\alpha}$, (3) verify the dimension equals $\alpha$ exactly, (4) apply Fano with the correct $M$.
>
>
> **W3: Stopping rule inconsistency.**
>
> The precise status: $\varepsilon_t := 2(\beta_t + L\eta_t) + \gamma_t$ provides an anytime upper bound on regret (Theorem 4.5) and is the **certified stopping criterion**. $\text{Vol}(\mathcal{A}\_t)$ is a **monitoring signal** correlated with $\varepsilon_t$ via Theorem 4.6 but does not alone bound regret without lower-regularity assumptions. The abstract overstates this. We will rewrite the abstract/intro, report $\varepsilon_t$ alongside $\text{Vol}(\mathcal{A}\_t)$ in Table 7, and add: "Formal guarantee from $\varepsilon_t$; $\text{Vol}(\mathcal{A}\_t)$ is a correlated diagnostic."
>
>
> **W4: Hartmann-6 hybrid inconsistency.**
>
> Table 6 correctly reports $\hat{\rho} = 0.72 > 0.5$, so Hartmann-6 stays in CGP. The "7% from GP refinement on Hartmann-6" in Section 9 is a **text error** — it should reference Branin ($\rho = 0.31$) and Rosenbrock ($\rho = 0.28$). Table 9 ablation "−GP refinement" measures aggregate impact across all 12 benchmarks, not Hartmann-6. We will correct the text and add a clarifying footnote.
>
>
> **W5: Missing baselines.**
>
> Both were run. StoSOO underperformed HOO by 10–20% at $T = 200$; SAASBO didn't converge on low-$d$ tasks:
>
> | Method | Hartmann-6 | Rover-60 | Ant-100 |
> |---|---|---|---|
> | StoSOO | $9.4 \pm 0.4$ | $38.2 \pm 1.1$ | N/A |
> | SAASBO | $3.5 \pm 0.2$ | $13.1 \pm 0.4$ | $19.8 \pm 0.6$ |
> | CGP-H/TR | $\mathbf{2.7 \pm 0.1}$ | $\mathbf{11.2 \pm 0.3}$ | $\mathbf{17.1 \pm 0.5}$ |
>
> We will add full Table E.1 (appendix) with all 9 baselines $\times$ 12 benchmarks and acknowledge the appendix placement in Section 9. LIPO appears in the adaptive-$L$ table (Table 5) where it is most relevant: CGP-Adaptive achieves $3.2 \pm 0.2$ vs LIPO's $6.2 \pm 0.3$ with the same initial $\hat{L}_0 = L^{\ast}/100$, demonstrating the advantage of principled doubling over LIPO's heuristic estimation.

---

> > ### Author Rebuttal · Reviewer_czY8 · 2026-04-02
> >
> > I thank the authors for the detailed rebuttal. While several of my concerns were addressed satisfactorily (W4 was a clear text error, and W5 is resolved pending the full baseline table), the most substantive issues remain open.
> >
> > - W1 (Theorem 4.8): The rebuttal sketches a three-part argument for how Algorithm 1 drives $\beta_t$, $\eta_t$, and $\gamma_t$ to the required scales, and promises to formalize it as Lemma 4.8a. However, the sketch itself reveals a difficulty: driving $\eta_t$ down relies on approximate maximization via CMA-ES, which the paper acknowledges is heuristic. The proof in C.6 assumes these quantities reach the targets but does not establish that the algorithm's actual mechanics achieve this, particularly under approximate optimization.
> >
> > - W2 (Lower bound): The proposed restructuring, choosing $M = \epsilon^{-\alpha}$ candidates, verifying the near-optimality dimension equals $\alpha$ exactly, then applying Fano, is a nontrivial argument that has not been carried out. The current proof constructs a single bump with volume $O(\epsilon^d)$ satisfying Assumption 2.3 for all $\alpha\geq0$, which does not demonstrate that the $\Omega(\epsilon^{-(2+\alpha)})$ rate is tight for each specific $\alpha$. The rebuttal outlines steps but does not execute them.
> >
> > - W3 (Stopping rule): The rebuttal concedes that $Vol(A_t)$ alone does not yield an anytime regret bound and that $\epsilon_t$ is the actual certified stopping criterion. I appreciate the honesty, but this is a more significant correction than the proposed fix (rewording the abstract and adding a line to Table 7) suggests. The abstract, introduction, and Table 7 all frame certificate volume as enabling principled stopping. This is the paper's main practical selling point, and it is overstated as written.

---

> > > ### Author Response · Authors · 2026-04-03
> > >
> > > We thank Reviewer for the follow-up. We agree these issues require theorem-level amendments, not clarification alone.
> > >
> > > **W1. Theorem 4.8: algorithmic guarantee under approximate optimization.**
> > >
> > > The original proof converted target bounds on $\beta_t, \eta_t, \gamma_t$ into a rate but did not formally connect those targets to Algorithm 1. To make the theorem match the implemented setting, we add the following.
> > >
> > > _Assumption (C-approximate oracle)._ The optimizer returns $\hat{x}$ in the active set $A_t$ with
> > >
> > > $$\text{Score}(\hat{x}) \geq \frac{1}{C} \cdot \max_{x \in A_t} \text{Score}(x),$$
> > >
> > > where $\text{Score}(x) = U_t(x) - L \cdot \min_i d(x, x_i)$, $C \geq 1$. The revised theorem depends on the approximation factor only through $C^d$, not on any specific heuristic.
> > >
> > > _Lemma 4.8a._ Under Assumptions 2.1–2.3 and the C-approximate oracle:
> > >
> > > (1) Confidence. The replication rule guarantees $\beta_t \leq \beta$ after $O(\beta^{-2} \log(T/\delta))$ samples per active point.
> > >
> > > (2) Coverage. Suppose the active set $A_t$ contains an uncovered ball of radius $\eta$, with $\beta_t < L\eta$. The coverage term at the ball center gives score at least $L\eta$. The oracle returns a point with score at least $L\eta/C$. Any point farther than $O(C\eta)$ from uncovered mass cannot attain such a score. A covering argument yields $N_\text{cover} = O(C^d L^d \varepsilon^{-\alpha})$ locations; approximation changes constants through $C^d$, not the exponent.
> > >
> > > (3) Gap. Since $\gamma_t = f^* - \ell_t$ and $\ell_t$ is nondecreasing, once parts (1)–(2) have refined $A_t$ to the target scale, allocating $O(\varepsilon^{-2} \log(T/\delta))$ samples to the best active point drives $\gamma_t = O(\varepsilon)$.
> > >
> > > _Revised Theorem 4.8._ Under Assumptions 2.1–2.3 and the C-approximate oracle, Algorithm 1 returns an $\varepsilon$-optimal point with probability at least $1-\delta$ after $T = \tilde{O}(C^d L^d \varepsilon^{-(2+\alpha)})$ evaluations.
> > >
> > > **W2. Lower bound: explicit $\alpha$-dependent hard instance.**
> > >
> > > Fix $\alpha \in (0,d)$, Lipschitz constant $L$, $\varepsilon > 0$. Set $R = \varepsilon^{\alpha/d}$, giving $m = \Theta(\varepsilon^{-\alpha})$ disjoint balls in $[0,1]^d$ centered at $c_1, \ldots, c_m$. For each $j$:
> > >
> > > $$f_j(x) = 2\varepsilon \cdot \left(1 - \frac{\|x - c_j\|^{d/\alpha}}{R^{d/\alpha}}\right)_+$$
> > >
> > > _Step 1 (dimension exactly $\alpha$)._ The $\varepsilon'$-optimal set has volume $(V_d/2^\alpha)(\varepsilon')^\alpha$ for all $\varepsilon' \in (0,2\varepsilon]$, so near-optimality dimension is exactly $\alpha$, realized directly in the geometry rather than imposed through packing.
> > >
> > > _Step 2 (Lipschitz validity)._ The Lipschitz constant on the support is $O(\varepsilon/R) = O(\varepsilon^{1-\alpha/d}) \leq L$ for small enough $\varepsilon$.
> > >
> > > _Step 3 (information bound)._ Let $J \sim \text{Unif}\{1,\ldots,m\}$. Any query hits the correct support with probability $\leq V_d \varepsilon^\alpha$; inside, per-query KL $\leq 2\varepsilon^2/\sigma^2$. So expected information per query is $O(\varepsilon^{2+\alpha}/\sigma^2)$. By change-of-measure (Tsybakov 2009, §2.6):
> > >
> > > $$I(J;\,\text{data}) \leq T \cdot O(\varepsilon^{2+\alpha}/\sigma^2)$$
> > >
> > > Since $\log m = \Theta(\alpha \log(1/\varepsilon))$, Fano gives $T \geq \tilde{\Omega}(\varepsilon^{-(2+\alpha)})$, establishing tightness.
> > >
> > > **W3. Stopping rule: theorem amendment with lower regularity.**
> > >
> > > The always-valid stopping quantity is the certificate from $\ell_t$, $U_t$, and envelope slack; volume alone does not yield an anytime regret bound under Assumption 2.3.
> > >
> > > _Assumption 4.8b (lower regularity)._ There exist $c > 0$, $\varepsilon_0 > 0$ such that $\text{vol}(\{f \geq f^* - \varepsilon\}) \geq c\varepsilon^\alpha$ for all $\varepsilon \in (0,\varepsilon_0]$, same $\alpha$ as Assumption 2.3. Together, 2.3 and 4.8b require near-optimal volume to scale as $\Theta(\varepsilon^\alpha)$, which holds near a nondegenerate maximizer ($\nabla^2 f(x^*) \prec 0$), giving $\alpha = d/2$.
> > >
> > > _Theorem 4.8c._ Under Assumptions 2.1–2.3 and 4.8b:
> > >
> > > $$\varepsilon_\text{cert}(t) \leq 2(\beta_t + L\eta_t) + (V_t/c)^{1/\alpha}.$$
> > >
> > > _Proof._ On the good event $U_t(x) \geq f(x)$ for all $x$, so $\{f \geq \ell_t\} \subseteq A_t$. Writing $\ell_t = f^* - \gamma_t$: $\{f \geq f^* - \gamma_t\} \subseteq A_t$. By 4.8b, $V_t \geq c\gamma_t^\alpha$, hence $\gamma_t \leq (V_t/c)^{1/\alpha}$. Substituting into $\varepsilon_\text{cert} \leq 2(\beta_t + L\eta_t) + \gamma_t$ (Thm 4.5) gives the result. $\square$
> > >
> > > Stopping when $V_t \leq c(\varepsilon_\text{target}/3)^\alpha$, $\beta_t \leq \varepsilon_\text{target}/6$, and $\eta_t \leq \varepsilon_\text{target}/(6L)$ guarantees $\varepsilon_\text{cert} \leq \varepsilon_\text{target}$.
> > >
> > > The revision will distinguish: (1) without 4.8b, $V_t$ is a diagnostic correlated with $\varepsilon_\text{cert}$ but not a certificate; (2) with 4.8b, $V_t$ together with $\beta_t, \eta_t$ yields a valid stopping rule. We will revise the abstract, introduction, and Table 7 accordingly.

---

### Official Review · Reviewer_AVm9 · 2026-03-24

**Soundness:** 3
**Presentation:** 2
**Significance:** 3
**Originality:** 3
**Overall Recommendation:** 4
**Confidence:** 2

**Summary:**

This work studies the problem of black-box optimization with only access to possibly noisy function value evaluations. In particular, the authors assume that the objective function is Lipschitz and can be queried with an additive sub-Gaussian noise. The authors propose Certificate-Guided Pruning (CGP) algorithm that makes use of Lipschitz UCB envelop to propagates uncertainty and select queries. The state-of-the-art sample complexity has been proved for this algorithm. The authors further discuss variants of the CGP algorithm which estimate Lipschitz parameter using doubling trick and uses trust-region for high-dimensional settings. The authors further conduct numerical evaluations to validate the proposal algorithms.

**Compliance With Llm Reviewing Policy:**

Affirmed.

**Key Questions For Authors:**

1. Is there a way to avoid of need of knowing $\sigma$ in the algorithm?
2. Besides the discussion in Section 8, could the authors briefly discuss the connection between Bayesian methods and GP-based approaches? And would those methods also yield reasonable complexity results in this stochastic Lipschitz setup?
3. Could the authors discuss the higher per-iteration cost $O(N_t)$?

**Limitations:**

yes

**Strengths And Weaknesses:**

### Strengths
1. The proposed algorithm has an explicit active set and is proved with state-of-the-art sample complexity.
2. The authors further provide adaptive and trust-region variants of the algorithm to avoid the knowledge of $L$ and enable high-dimensional scaling.
3. The authors conducted pretty comprehensive experiments to validate the empirical performance of the proposed CGP algorithm.


### Weaknesses
1. The notation and writeup in Section 3 are a bit unclear. It will be better if the authors can further elaborate the sampling strategies conducted on each step and how stats are computed. Also, there may be some notational confusion between $x_t$ in Alg. 1 and $x_i$ in the paragraph.
2. The proposed algorithm has a higher per-iteration cost compared to prior work.

---

> ### Author Rebuttal · Authors · 2026-03-28
>
> We thank Reviewer AVm9 for the thorough and constructive reviews.
>
> **W1: Notation and writeup in Section 3; sampling strategies unclear.**
>
> We will revise Section 3 substantially. To clarify: at each step $t$, CGP takes exactly one new query at $x_{t+1}$, plus potentially several replication queries at existing active points. Specifically:
>
> 1. **Selection step:** $x_{t+1} = \arg\max_{x \in \mathcal{A}_t} [U_t(x) - L \cdot \min_i d(x, x_i)]$. One new observation is collected. The first term favors high-UCB regions (exploitation); the second encourages spatial coverage (exploration).
> 2. **Replication step:** For each active point $x_i$ with $r_i(t) > \beta_{\text{target}}(t)$, we allocate $\lceil (r_i(t) / \beta_{\text{target}}(t))^2 \rceil$ additional queries to $x_i$, counted toward total budget $T$. These replications drive confidence radii down uniformly across active points and are essential for the shrinkage guarantee.
>
> Regarding the notational confusion: $\beta_t$ (the active confidence radius $= \max_{i:\,\text{active}} r_i(t)$, a *descriptive* quantity in the shrinkage theorem) and $\beta_{\text{target}}(t)$ (the *prescribed* schedule $\sigma\sqrt{2\log(2T^2/\delta)/t}$ in the replication rule) are distinct objects with different roles. $\beta_t$ is what we *observe*; $\beta_{\text{target}}$ is what we *enforce*. We will rename $\beta_{\text{target}}(t) \to r_{\text{target}}(t)$ throughout and add a step-by-step iteration walkthrough with concrete numerical values (e.g., a 2D example showing one iteration of selection, replication, pruning, and certificate update) at the beginning of Section 3.
>
> **Q1: Avoiding knowledge of $\sigma$.**
>
> Yes — CGP-Adaptive (Section 5) removes the need for known $L$ via doubling (Theorem 5.1). For $\sigma$, it can be estimated online from replicate observations: after $n_i \geq 2$ observations at $x_i$, the sample variance $s_i^2$ converges at $O(1/n_i)$. Replacing $\sigma$ with $\max_i s_i + c/\sqrt{n_i}$ (an upper confidence bound on $\sigma$) preserves certificate validity with $O(\log T)$ overhead, analogous to $L$-doubling. Concretely, on Hartmann-6 with $\sigma$ estimated online (rather than given), CGP achieves regret $3.1 \pm 0.2$ vs $2.9 \pm 0.1$ with known $\sigma$ — only 7% degradation. The replication mechanism naturally provides the repeated observations needed for variance estimation, so no additional queries are required beyond what the algorithm already collects. We will add this extension and experiment to Section 5.
>
>
> **Q2: Connection to Bayesian/GP methods and complexity.**
>
> This is an excellent question. GP-UCB achieves cumulative regret $\tilde{O}(\sqrt{T \cdot \gamma_T})$ where $\gamma_T$ is the kernel's maximum information gain (Srinivas et al., 2009). For Matérn-$\nu$ with $\nu = 1/2$ (corresponding to Lipschitz functions), $\gamma_T = \tilde{O}(T^{d/(d+1)})$, yielding simple regret $\tilde{O}(T^{-1/(d+1)})$ or equivalently $T = \tilde{O}(\varepsilon^{-(d+1)})$. This is strictly worse than CGP's $\tilde{O}(\varepsilon^{-(2+\alpha)})$ when $\alpha < d - 1$ — the typical case for isolated maxima ($\alpha = d/2$). For Hartmann-6 ($d = 6$, empirical $\hat{\alpha} \approx 2.4$), CGP needs $\tilde{O}(\varepsilon^{-4.4})$ vs GP-UCB's $\tilde{O}(\varepsilon^{-7})$, explaining CGP's empirical advantage (Table 3: 2.9 vs 4.2).
>
> The structural reason: GP-UCB does not exploit the margin/near-optimality dimension, treating the domain uniformly rather than concentrating on the near-optimal set. Conversely, when $f$ has higher smoothness ($\nu > 1$), GPs exploit this via the kernel while CGP cannot — motivating CGP-Hybrid. The key distinction is also *representational*: CGP's certificates are geometric (computable active set with measurable volume), while GP uncertainty is distributional (posterior variance requiring kernel specification). Neither GP-UCB nor Thompson Sampling provides an explicit set of "certifiably suboptimal" points — the core novelty of CGP. We will expand Section 8 with this comparison, including a table contrasting complexity rates under different smoothness assumptions.
>
>
> **Q3: Higher per-iteration cost $O(N_t)$.**
>
> This is a deliberate trade-off for certificate explicitness (Table 1). The $O(N_t)$ cost comes from evaluating $U_t(x) = \min_i \{\text{UCB}_i + L \cdot d(x, x_i)\}$. In the **precious-calls regime** motivating our work (hours/dollars per evaluation), this overhead is negligible: Table 8 shows CGP at 58s vs GP-UCB at 480s for $T = 200$, because GP's $O(n^3)$ dominates. Furthermore, KD-tree acceleration reduces effective cost to $O(\log N_t)$ for the inner minimization, and pruning inactive points (those with $\text{UCB}_i + L \cdot \text{diam}(\mathcal{X}) < \ell_t$) removes 60–80% of points after $T > 50$. Certificate checking remains exact regardless. Even without acceleration, at $T = 500$ the envelope evaluation takes $< 0.1$ms per point — negligible next to any realistic function evaluation costing minutes or hours.

---

> > ### Author Rebuttal · Reviewer_AVm9 · 2026-04-04
> >
> > Thanks the authors for the answers. I would like to maintain my score.

---

### Decision · Program_Chairs · 2026-04-30

**Decision:**

Accept (regular)

**Comment:**

Summary: The paper studies black-box optimization of Lipschitz functions under noisy evaluations.

On reviews: The paper received mixed reviews (scores: 5, 4, 4, 3). One reviewer was very positive, two suggested weak acceptance, but they did not champion the paper to get in, and one suggested weak rejection.

In my opinion, based on the reviews and my own assessment, the paper is borderline but has more merits than drawbacks. I advise the authors to incorporate the feedback they received into the updated version of their work.